# Towards Empirical Interpretation of Internal Circuits and Properties in Grokked Transformers on Modular Polynomials

**Hiroki Furuta    Gouki Minegishi    Yusuke Iwasawa    Yutaka Matsuo**
*{furuta,minegishi,iwasawa,matsuo}@weblab.t.u-tokyo.ac.jp*
*The University of Tokyo*

**Reviewed on OpenReview:** *https://openreview.net/forum?id=MzSf7OuXJO*

## Abstract

Grokking has been actively explored to reveal the mystery of delayed generalization and identifying interpretable representations and algorithms inside the grokked models is a suggestive hint to understanding its mechanism. Grokking on modular addition has been known to implement Fourier representation and its calculation circuits with trigonometric identities in Transformers. Considering the periodicity in modular arithmetic, the natural question is to what extent these explanations and interpretations hold for the grokking on other modular operations beyond addition. For a closer look, we first hypothesize that (1) any modular operations can be characterized with distinctive Fourier representation or internal circuits, (2) grokked models obtain common features transferable among similar operations, and (3) mixing datasets with similar operations promotes grokking. Then, we extensively examine them by learning Transformers on complex modular arithmetic tasks, including polynomials. Our Fourier analysis and novel progress measure for modular arithmetic, *Fourier Frequency Density* and *Fourier Coefficient Ratio*, characterize distinctive internal representations of grokked models per modular operation; for instance, polynomials often result in the superposition of the Fourier components seen in elementary arithmetic, but clear patterns do not emerge in challenging non-factorizable polynomials. In contrast, our ablation study on the pre-grokked models reveals that the transferability among the models grokked with each operation can be only limited to specific combinations, such as from elementary arithmetic to linear expressions. Moreover, some multi-task mixtures may lead to co-grokking – where grokking simultaneously happens for all the tasks – and accelerate generalization, while others may not find optimal solutions. We empirically provide significant steps towards the interpretability of internal circuits learned through modular operations, where analytical solutions are not attainable.

## 1 Introduction

Grokking is a late generalization phenomenon when training Transformer (Vaswani et al., 2017) and other architectures with algorithmic data (Power et al., 2022) where training accuracy soon reaches 100% with low test accuracy (often 0%), and after several iterations, test accuracy gradually reaches 100%. Grokking has been actively explored to reveal the mystery of delayed generalization, and identifying interpretable circuits inside the grokked models should be a suggestive hint to understanding the grokking mechanism and dynamics. The interpretability analysis has mainly shed light on modular addition, where grokking obtains the calculation with Fourier basis and trigonometric identities (Nanda et al., 2023; Zhong et al., 2023; Gromov, 2023; Rubin et al., 2023). Considering the periodicity in modular arithmetic, the natural question is to what extent these explanations and interpretations hold for the grokking on other modular operations beyond addition.

For a closer look at the connections among the grokking phenomena in other modular operations, we first hypothesize that (1) *any modular operations can be characterized with unique Fourier representation or*

*algorithms* (**circuit formulation**), (2) *grokked models obtain common features transferable among similar operations* (**transferability**), and (3) *mixing functionally similar operations in dataset promote grokking* (**multi-task training**). Revealing these relations would help us understand and analyze the dynamics of grokking better. In this work, beyond the simplest and well-studied operation, we observe the internal circuits learned through grokking in complex modular arithmetic via interpretable reverse engineering, and extensively verify our three hypotheses, while also investigating whether grokked models may exhibit transferability among the models grokked with other operations and scaling to the similarity and the number of tasks[1].

First, analyzing modular subtraction, multiplication, and polynomials reveals that the operations that cause grokking have unique Fourier representations (Section 5). For instance, subtraction poses a strong asymmetry on Transformer (Section 5.1), and multiplication requires cosine-biased components at all frequencies (Section 5.2). Grokking can easily occur in certain modular polynomials, such as the sum of powers and higher-degree expressions factorizable with basic symmetric and alternating expressions (Section 6). These polynomials have a superposition of representations in modular elementary arithmetic, while "non-grokked" operations do not have explicit patterns (Section 6.1). We also introduce the novel progress measure for modular arithmetic; *Fourier Frequency Density (FFD)* and *Fourier Coefficient Ratio (FCR)*, which not only indicate the late generalization but also characterize distinctive internal representations of grokked models per modular operation (Section 6.3). We prove that our proposed FFD and FCR decrease accompanying the test accuracy improvement, and they reflect features of internal circuits, such as the coexistence of addition and multiplication patterns in $ab + b$, or dependence of the factorizable polynomials on the parity of exponent $n$.

In contrast, the ablation study with pre-grokked models reveals that the transferability of grokked embeddings and models is limited to specific combinations, such as from elementary arithmetic to linear expressions (Section 7.1), and could be rarely observed in higher-degree expressions (Section 7.2). Besides, some mixtures of multiple operations lead to the co-occurrence of grokking and even accelerate generalization (Section 8.1). In contrast, others may interfere with each other, not reaching optimal solutions (Section 8.2). These observations indicate that the mechanism of grokking might not always share the underlying dynamics with common machine learning. We provide significant insights in the empirical interpretation of internal circuits learned through modular operations, where analytical solutions are not attainable.

## 2 Related Work

**Grokking**  Grokking has been actively studied to answer the questions: (1) when it happens, (2) why it happens, and (3) what representations are learned. In simple algorithmic tasks like modular addition, grokking would be observed with proper weight decay and the ratio of train-test splits (Power et al., 2022; Lyu et al., 2023). In addition to synthetic data (Liu et al., 2023b), grokking could occur in more general settings such as teacher-student (Levi et al., 2023), NLP (Murty et al., 2023), computer vision (Thilak et al., 2022), or molecular graph tasks (Liu et al., 2023a), which could be explained with the dynamic phase transitions during training (Rubin et al., 2023; Kumar et al., 2023) or mismatch between the train-test loss landscape against weight norm (Liu et al., 2023a). Recent findings have revealed that while grokking has initially been observed in neural networks (MLP and Transformer) with weight decay, it may also occur in Gaussian processes and linear regression models (Levi et al., 2023; Miller et al., 2023) or even the case without weight decay (Kumar et al., 2023). Our work focuses on complex modular arithmetic including subtraction, multiplication, polynomials, and a multi-task mixture, and then empirically analyzes the difference between grokked and non-grokked modular operations.

Several works have argued that the late generalization dynamics has been driven by the sparsification of neural network emerging dominant sub-networks (Merrill et al., 2023; Tan & Huang, 2023) and the structured representations (Liu et al., 2022); the training process could be a phase transition divided into memorization, circuit formation, and cleanup phase (Nanda et al., 2023; Xu et al., 2023; Doshi et al., 2023; Davies et al., 2023; Žunkovič & Ilievski, 2022), and the formation of generalization circuits produces higher logits with small norm parameters than memorization circuits (Varma et al., 2023). The sparse lottery tickets in neural

---

[1]https://github.com/frt03/grok_mod_poly

networks may also promote grokking (Minegishi et al., 2023). Moreover, our work highlights that in modular arithmetic such sparse representations are obtained interpretably through the discrete Fourier transform.

**Mechanistic Interpretability** While training neural networks is often accompanied by mysterious phenomena such as double descent (Nakkiran et al., 2019), many works along the mechanistic interpretability have attempted to systematically understand what happened during training and inference through extensive reverse engineering (Olah et al., 2020; Olsson et al., 2022; Akyürek et al., 2023; Elhage et al., 2022; Notsawo et al., 2023). Paying attention to the activation of neurons, those studies have tried to identify the functional modules or circuits inside neural networks (Elhage et al., 2021; Conmy et al., 2023). Even for recent large language models, controlling activation patterns via activation patching can unveil the role of each module (Vig et al., 2020; Meng et al., 2023; Zhang & Nanda, 2024). In grokking literature, several works have revealed what kind of algorithmic pattern was obtained inside the model when it worked on modular addition (Zhong et al., 2023; Nanda et al., 2023; Morwani et al., 2023) or group composition (Chughtai et al., 2023; Stander et al., 2023) through the Fourier transform of logits or investigating gradients. Gromov (2023) points out that the learned weights and algorithms in some arithmetic tasks are analytically solvable if MLP uses a quadratic activation. In contrast, we extend the analysis from addition to other modular operations, such as subtraction, multiplication, polynomials, and a multi-task mixture, which can bridge the gap between simple synthetic data from modular addition and complex structured data as seen in the real world.

## 3 Preliminaries

**Grokking** This paper focuses on grokking on the classification tasks from simple algorithmic data commonly investigated in the literature (Power et al., 2022; Liu et al., 2022; Barak et al., 2022). We have train and test datasets ($\boldsymbol{S}_{\text{train}}$, $\boldsymbol{S}_{\text{test}}$) without overlap, and learn a neural network $f(\boldsymbol{x}; \boldsymbol{\theta})$ where input $\boldsymbol{x}$ is a feature vector of elements in the underlying algorithm space for synthetic data and $\boldsymbol{\theta}$ are weights of neural network. The small-size Transformers (e.g. one or two layers) or MLP are usually adopted as $f$. Specifically, prior works train the network using stochastic gradient decent over the cross-entropy loss $\mathcal{L}$ and weight decay:

$$\boldsymbol{\theta} \leftarrow \underset{\boldsymbol{\theta}}{\operatorname{argmin}} \ \mathbb{E}_{(\boldsymbol{x}, y) \sim \boldsymbol{S}} \left[ \mathcal{L}(f(\boldsymbol{x}; \boldsymbol{\theta}), y) + \frac{\lambda}{2} \|\boldsymbol{\theta}\|_2 \right],$$

where $y \in \{0, ..., p-1\}$ is a scalar class label ($p$ is a number of classes) correspond to the inputs $\boldsymbol{x} \in \{0, ..., p-1\} \times \{0, ..., p-1\}$, and $\lambda$ is a hyper-parameter controlling the regularization. The weight decay is one of the factors inducing the grokking phenomenon (Power et al., 2022; Liu et al., 2023a; Varma et al., 2023). We employ AdamW (Loshchilov & Hutter, 2019)) as an optimizer in practice. The fraction of training data from all the combinations is defined as:

$$r = \frac{|\boldsymbol{S}_{\text{train}}|}{|\boldsymbol{S}_{\text{train}}| + |\boldsymbol{S}_{\text{test}}|} \left( = \frac{|\boldsymbol{S}_{\text{train}}|}{p^2} \right).$$

It has been observed that a larger fraction tends to facilitate fast model grokking, whereas a smaller fraction makes grokking more challenging and slow especially in complex settings such as modular polynomial tasks.

**Transformers** As discussed in Elhage et al. (2021), the functionality of a small-size Transformer can be written down with several distinctive matrices. We denote embedding weights as $W_E \in \mathbb{R}^{d_{\text{emb}} \times p}$, output weights at the last MLP block as $W_{\text{out}} \in \mathbb{R}^{d_{\text{emb}} \times d_{\text{mlp}}}$, and unembedding weights as $W_U \in \mathbb{R}^{p \times d_{\text{emb}}}$. The logit vector on inputs $a, b$ can be approximately written with activatations from MLP block, $\text{MLP}(a, b)$, as $\text{Logits}(a, b) \approx W_U W_{\text{out}} \text{MLP}(a, b)$ by ignoring residual connection (Nanda et al., 2023), and we investigate the neuron-logit map $W_L = W_U W_{\text{out}} \in \mathbb{R}^{p \times d_{\text{mlp}}}$ in the later analysis. See Appendix A for further details.

**Analysis in Modular Addition** Nanda et al. (2023) first pointed out that Transformer uses particular Fourier components and trigonometric identities after grokking occurred in modular addition [2]. The modular addition is a basic mathematical operation as $(a + b) \% p = c$ where $a, b, c$ are integers. The model predicts $c$ given a pair of $a$ and $b$. As a slightly abused notation, $a, b, c$ may represent one-hot representation, and we

---

[2]Zhong et al. (2023) revealed the existence of algorithms independent of trigonometric identities if the model does not have attention. Since we employ a Transformer with attention, this paper assumes the algorithms based on trigonometric identities.

will omit $\% \ p$ in later sections. In the case of modular addition, the way Transformer model represents the task has been well-studied (Zhong et al., 2023; Nanda et al., 2023), where the embedding matrix $W_E$ maps the input one-hot vectors into cosine and sine functions for various frequencies $\omega_k = \frac{2k\pi}{p}, k \in \{0, ..., p-1\}$, such as $a \to \cos(\omega_k a), \sin(\omega_k a)$. It is also known that the addition is implemented inside the Transformer with trigonometric identities,

$$\cos(\omega_k(a+b)) = \cos(\omega_k a)\cos(\omega_k b) - \sin(\omega_k a)\sin(\omega_k b),$$
$$\sin(\omega_k(a+b)) = \sin(\omega_k a)\cos(\omega_k b) + \cos(\omega_k a)\sin(\omega_k b),$$

and then the neuron-logit map $W_L$ reads off $\cos(\omega_k(a+b-c))$ by also using trigonometric identities,

$$\cos(\omega_k(a+b-c)) = \cos(\omega_k(a+b))\cos(\omega_k c) + \sin(\omega_k(a+b))\sin(\omega_k c). \tag{1}$$

The logits of $c$ are the weighted sum of $\cos(\omega_k(a+b-c))$ over $k$. Note that we only consider the first half of frequencies (i.e. $k \in \{1, ..., [\frac{p}{2}]\}$) because of the symmetry. We show the example Python code for Fourier analysis in Appendix B.

**Experimental Setup** In this paper, we expand the discussion above on modular addition to entire modular arithmetic: $a \circ b \ \% \ p = c$ where $\circ$ represents arbitrary operations (or polynomials) that take two integers $a$ and $b$ as inputs, such as $a - b$ (subtract), $a * b$ (multiplication), $2a - b$, $ab + b, a^2 + b^2, a^3 + ab, (a+b)^4$ (polynomials) [3]. Transformer takes three one-hot tokens as inputs, $a$, $\circ$, $b$. In addition to $p$ integer tokens, we prepare $n_{\text{op}}$ special tokens representing the mathematical operations above. The models are trained to predict $c$ as an output.

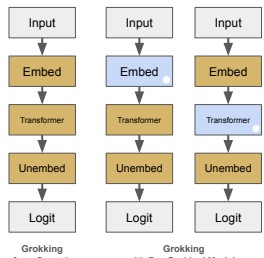

Our neural network is composed of a single-layer causal Transformer architecture (Figure 1) with learnable embedding and unembedding ($d_{\text{emb}} = 128$). We use ReLU for the activation functions and remove positional embedding, layer normalization, and bias terms for all the layers. We initialize each weight from Gaussian distribution, where the mean is 0 and the standard deviation is $\frac{1}{\sqrt{d_{\text{out}}}}$ (LeCun et al., 2012). This Transformer is trained via full batch gradient descent with AdamW (Loshchilov & Hutter, 2019) and weight decay $\lambda = 1.0$. We use $p = 97$ for all the experiments. For the dataset faction, we use $r = 0.3$ unless otherwise mentioned. We conduct the experiments with 3 random seeds and report the average[4]. Other hyper-parameters are described in Appendix C.

Figure 1: Grokking has been investigated with training from scratch. To shed light on the dynamics inside Transformer, we introduce the notion of *pre-grokked models*, which are pre-trained on a similar task until grokking and used to replace randomly initialized modules without any parameter updates (i.e. frozen). We use pre-grokked embedding and Transformer in the later section.

## 4 Pre-Grokked Models and Fourier Metrics

In contrast to modular addition, the exact analysis of internal circuits across other modular arithmetic would be challenging, since not all the operations have analytical algorithms. To mitigate such interpretability issues, we introduce the notion of *pre-grokked models*, and propose a pair of novel progress measures for grokking in modular arithmetic; *Fourier Frequency Density (FFD)* and *Fourier Coefficient Ratio (FCR)*, which are derived from our empirical observation on sparsity and sinusoidal bias in embedding and neuron-logit map.

**Pre-Grokked Models** To dive into the internal dynamics, we leverage pre-grokked models, which are pre-trained on similar algorithmic tasks until grokking and used for another training to replace randomly initialized modules without any parameter updates (i.e. frozen). This allows us to consider learning representations and algorithms separately. We will use pre-grokked embedding (freezing $W_{\text{emb}}$) and Transformer (freezing all the weights except for $W_{\text{emb}} and W_{\text{U}}$) in later sections. We also use the same random seed between pre-grokking and downstream grokking experiments to reduce identifiability issues (Singh et al., 2024).

---

[3] We omit the discussion on modular division, since it requires division into cases while we also consider a multi-task mixture.
[4] We optimize the models up to 3e5 gradient steps and then judge whether grokking happens or not.

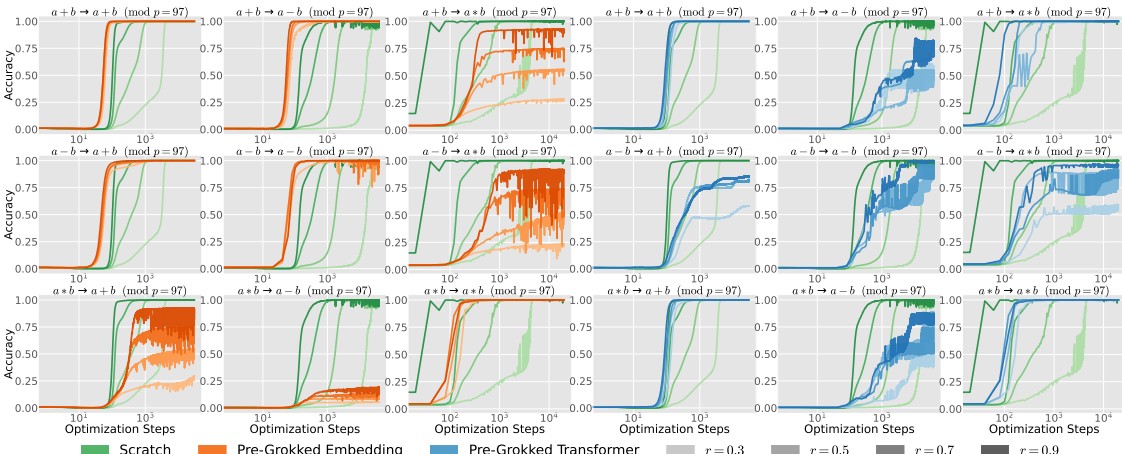

Figure 2: Test accuracy in modular elementary arithmetic (addition, subtraction, and multiplication) with pre-grokked models (embedding and Transformer). The x-axis is the logarithmic scale. Because of the task simplicity, grokking always occurs in elementary arithmetic. However, in certain combinations, pre-grokked models hinder grokking even with a $r = 0.9$ fraction. For pre-grokked embedding, addition and subtraction accelerate grokking each other (fig[0:2, 0:2]), while multiplication and those do not show synergy (+: fig[2, 0] and [0, 2], −: fig[2, 1] and [1, 2]). In contrast, for pre-grokked Transformer, subtraction is challenging in both directions, even transferring subtraction models into subtraction itself (fig[1, 4]). With small $r$, addition and multiplication accelerate each other (fig[0, 5] and [2, 3]).

**Fourier Frequency Density (FFD)** FFD quantitatively measures the sparsity of Fourier components in a certain layer (embedding or neuron-logit map),

$$\mathrm{FFD}(\eta, \boldsymbol{\mu}, \boldsymbol{\nu}) = \frac{1}{2\left[\frac{p}{2}\right]} \sum_k^{\left[\frac{p}{2}\right]} \mathbb{1}\left[\frac{\|\mu_k\|_2}{\max_{\boldsymbol{\mu}}\|\mu_i\|_2} > \eta\right] + \mathbb{1}\left[\frac{\|\nu_k\|_2}{\max_{\boldsymbol{\nu}}\|\nu_j\|_2} > \eta\right],$$

where $u_k \in \boldsymbol{\mu} = \{\mu_1, ..., \mu_k, ...\}$ is a coefficient of cosine components ($\mu_k = \frac{2}{\left[\frac{p}{2}\right]} \sum_{t=1}^{\left[\frac{p}{2}\right]} W[t] \cos(\omega_k t)$; $W[t]$ is a $t$-th index of weight matrix $W$) and $\nu_k \in \boldsymbol{\nu}$ is a coefficient of sine components ($\nu_k = \frac{2}{\left[\frac{p}{2}\right]} \sum_{t=1}^{\left[\frac{p}{2}\right]} W[t] \sin(\omega_k t)$) with frequency $\omega_k$. We set $\eta = 0.5$. The low FFD indicates that a few key frequencies are dominant in the Fourier domain, which can be often observed in modular addition.

**Fourier Coefficient Ratio (FCR)** FCR quantifies the sinusoidal bias of Fourier components in a certain weight matrix,

$$\mathrm{FCR}(\boldsymbol{\mu}, \boldsymbol{\nu}) = \frac{1}{\left[\frac{p}{2}\right]} \sum_k^{\left[\frac{p}{2}\right]} \min\left(\frac{\|\mu_k\|_2}{\|\nu_k\|_2}, \frac{\|\nu_k\|_2}{\|\mu_k\|_2}\right).$$

The low FCR means that Fourier representations of the weights have either cosine- or sine-biased components, which can be often observed in modular multiplication.

The decrease of either FFD or FCR (or both) correlates to the progress of grokking, and the responsible indicator varies for each modular operation; for instance, FFD is a good measure for addition, and FCR is for multiplication. They are not only aligned with the late improvement in test accuracy but also can characterize each Fourier representation of modular operations at a certain layer (Section 6.3).

## 5 Analysis in Elementary Arithmetic

We start with the analysis with internal circuits of pre-grokked models, which can reveal the characteristics of each arithmetic operation; if pre-grokked embedding encourages grokking in downstream tasks, the learned embedding should be similar, but if pre-grokked embedding prevents grokking, those tasks may require

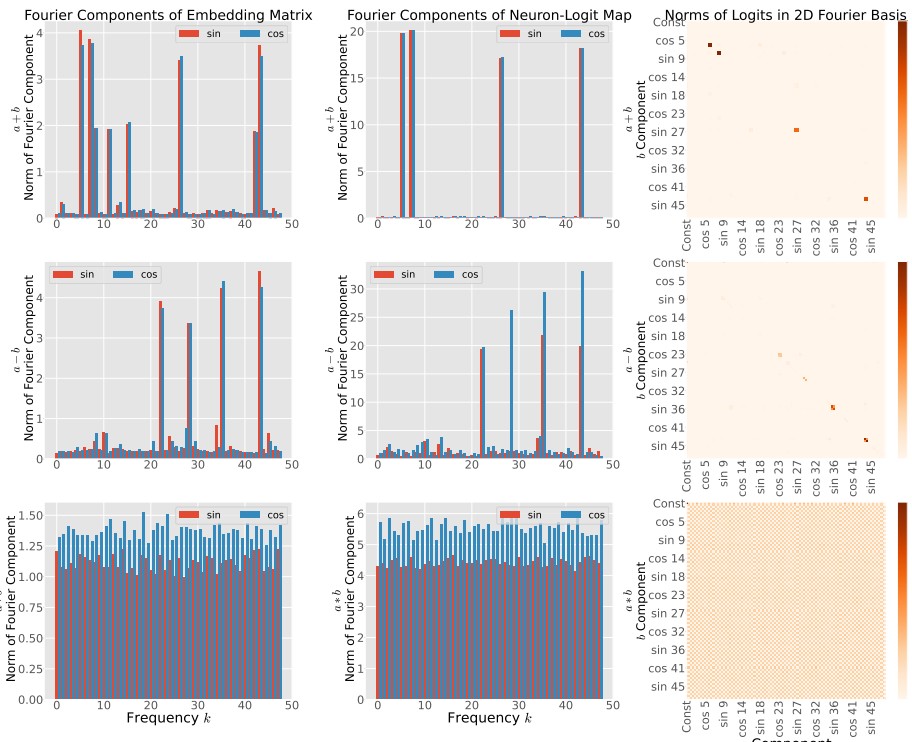

Figure 3: Frequency analysis in grokking with elementary arithmetic. Subtraction learns similar embedding to addition with sparse Fourier components (fig[0, 0] and fig[1, 0]). However, it imposes an asymmetric neuron-logit map and norm of logits with cosine biases (fig[1, 1] and fig[1, 2]). Multiplication obtains quite a different embedding from others (fig[2, :]); it employs all the frequencies equally with cosine bias for both embedding and neuron-logit map.

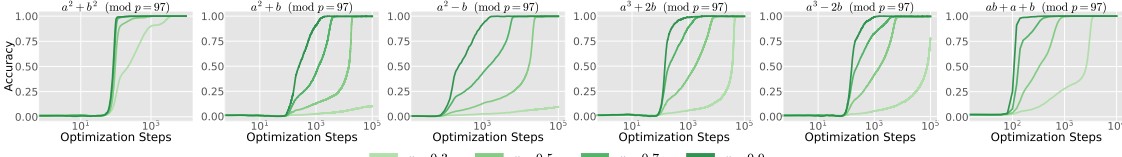

Figure 4: Test accuracy in modular polynomials (univariate terms: $a^2 + b^2$, $a^2 \pm b$, $a^3 \pm 2b$, the degree-1 with cross term: $ab + a + b$). Grokking occurs even in quadratic or cubic expressions asymmetric with input $a$ and $b$.

different types of representations. Moreover, if a pre-grokked Transformer accelerates generalization, it means the algorithms obtained internally would have similar properties, while the failure may hint at algorithmic differences.

Figure 2 shows test accuracy in elementary arithmetic (addition, subtraction, and multiplication) with pre-grokked embedding and Transformer[5]. Because of the task simplicity, grokking always occurs among those operations. However, in certain combinations, pre-grokked models hinder grokking even with a $r = 0.9$ fraction. For pre-grokked embedding, modular addition and subtraction accelerate grokking (Figure 2[0:2, 0:2]), while modular multiplication and those two hurt the performances each other (+: Figure 2[2, 0] and [0, 2], −: Figure 2[2, 1] and [1, 2]). In contrast, for pre-grokked Transformer, modular subtraction is challenging in both directions, even transferring subtraction models into subtraction itself (Figure 2[1, 4]). Pre-grokked Transformer on addition or multiplication accelerate each other (Figure 2[0, 5] and [2, 3]). Those results might imply that (1) while there is a similarity between the learned embeddings in addition and subtraction, their acquired algorithms significantly differ (Section 5.1), and that (2) multiplication requires representations independent of addition or subtraction but the algorithm might be transferable (Section 5.2).

---

[5]To avoid the confusion, we will mention the sub-figures using pythonic coordinates like Figure[i, j] for row i column j.

### 5.1 Modular Subtraction Imposes Strong Asymmetry

Considering the sign in trigonometric identities, Transformers should learn modular subtraction in the Fourier domain with trigonometric identities as the case of addition (Equation 1):

$$\cos(\omega_k(a - b - c)) = \cos(\omega_k(a - b))\cos(\omega_k c) + \sin(\omega_k(a - b))\sin(\omega_k c),$$

and then we would anticipate similar interpretable representations to addition. However, we observe that the grokked models exhibit asymmetric properties for both embedding and Transformer. We transform the embedding into a Fourier domain along the input dimension and compute the L2 norm along other dimensions. In Figure 3, subtraction learns similar embedding to addition with sparse Fourier components (Figure 3[0, 0] and [1, 0]). On the other hand, it imposes an asymmetric neuron-logit map and norms of logits with cosine-biased components (Figure 3[1, 1] and [1, 2]), which may represent alternatings $(a - b \neq b - a)$.

Such an asymmetry is also observed in grokked Transformers. The pre-grokked Transformer on subtraction could not be transferred to **any** downstream elementary arithmetic (Figure 2[1, :]), even subtraction itself (Figure 2[1, 4]), and pre-grokked models with addition or multiplication could not learn subtraction as well (Figure 2[:, 4]). This implies that while we could interpret subtraction as a part of addition with negative numbers, it is possible that the embedding and algorithm inside Transformer are quite different.

Lastly, we examine the restricted loss and ablated loss in Appendix E, where the restricted loss is calculated only with the Fourier components of significant frequencies, and the ablated loss is calculated by removing a certain frequency from the logits. The analysis emphasizes the subtle dependency on other frequencies than significant ones.

### 5.2 Modular Multiplication Leverages All Frequencies

In contrast to modular addition and subtraction, we do not attempt to fully interpret the learned model in a closed form or a form of pseudocode. However, following the empirical analysis in modular addition, we can observe that multiplication also leverages the periodicity in the Fourier domain.

Figure 3 reveals that multiplication obtains significantly different Fourier representation from addition or subtraction (Figure 3[2, :]); it employs all the frequencies equally with cosine bias for both embedding and neuron-logit map. Surprisingly, multiplication-pre-grokked Transformer accelerates grokking in addition (Figure 2[2, 3]) and addition-pre-grokked Transformer (Figure 2[0, 5]) causes grokking in multiplication. This implies that in contrast to the asymmetry of subtraction, addition, and multiplication leverage their symmetry in the operations. Since the embedding of multiplication is quite different from addition and subtraction, it seems to be reasonable to fail to grok with addition/subtraction-pre-grokked embeddings (Figure 2[0:2, 2] and [2, 0:2]). Moreover, we find that grokking in elementary arithmetic occurs even with frozen random embedding (see Appendix F; sampled from the same random seed) that does not have biased components nor sparsity, which also supports that some unique, non-transferable patterns are learned in grokked models.

## 6 Analysis in Polynomials

It has been known that grokking would be less likely to occur as increasing the complexity of operators in general (Power et al., 2022), but the underlying reasons or conditions are still unclear. In addition to elementary operations, we examine the interpretable patterns of grokked models in modular polynomials. We first investigate the case of simple polynomials (Section 6.1), quadratic, cubic, and quartic expressions (Section 6.2).

### 6.1 Polynomials Discover Superposition of Representations for Elementary Arithmetic

We here investigate the relatively simple polynomials that induce grokking (univariate terms: $a^2 + b^2$, $a^2 \pm b$, $a^3 \pm 2b$, the degree-1 with cross term: $ab + a + b$). In Figure 4, grokking occurs even in quadratic or cubic expressions asymmetric with input $a$ and $b$, and suggests that the existence of symmetry or the cross term might be a key for occurrence.

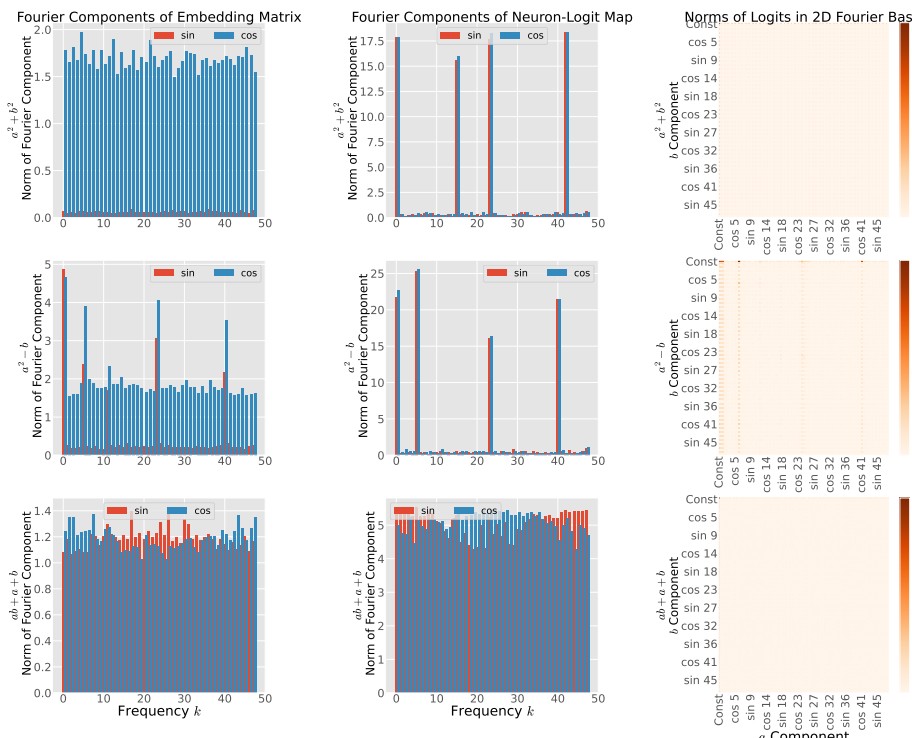

Figure 5: Frequency analysis in grokking with modular polynomials ($a^2 + b^2$, $a^2 - b$, $ab + a + b$). Grokking discovers the superposition of frequency sparsity and bias seen in elementary arithmetic; $a^2 - b$ inherits both biased sparsity in subtraction and significant cosine biases in multiplication for embedding (fig[1,0]). Its neuron-logit map leverages addition-like sparsity (fig[1,1]).

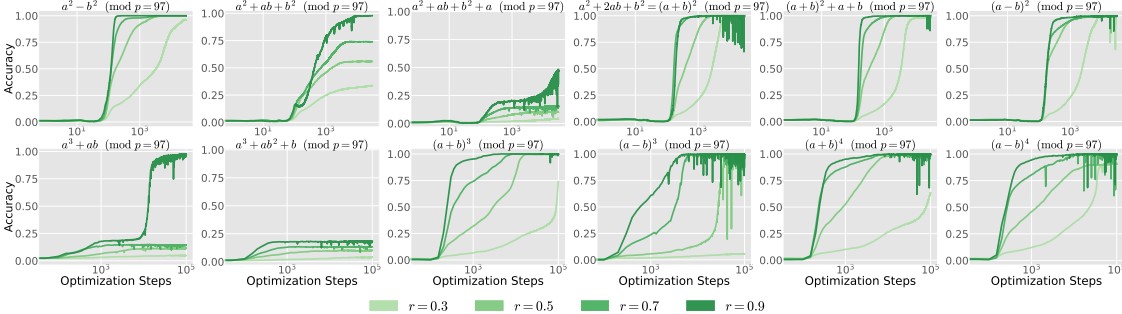

Figure 6: Test accuracy in modular polynomials with quadratic, cubic, and quartic formulas. Compared to the elementary arithmetic (Figure 2), these requires longer optimization steps. Transformers suffer from late generalization in degree-$n$ polynomials with cross-term ($a^2 + ab + b^2$, $a^2 + ab + b^2 + a$, $a^3 + ab$, $a^3 + ab^2 + b$). If polynomials are factorizable with addition ($a + b$) or subtraction ($a - b$), they are easy to grok (e.g. $(a + b)^2 + a + b$; fig[0][4]) although they also have a cross term (c.f. $a^2 + ab + b^2$). Even, cubic ($(a \pm b)^3$; fig[1][2:4]) or quartic ($(a \pm b)^4$; fig[1][4:]) expressions, grokking occurs if they are factorizable.

Moreover, the grokked models exhibit partially-similar internal states to the one in elementary arithmetic. Figure 5 provides frequency analysis with modular polynomials ($a^2 + b^2$, $a^2 - b$, $ab + a + b$), where grokking discovers superposition of representations (frequency sparsity and bias) for elementary arithmetic. For instance, $a^2 + b^2$ finds a cosine-biased embedding like multiplication and a sparse neuron-logit map like addition. $a^2 - b$ inherits both biased sparsity in subtraction and significant cosine biases in multiplication for embedding. Its neuron-logit map leverages addition-like sparsity. $ab + a + b$ is similar to multiplication; leveraging biased all the frequencies while using sine components, because it can be factorized as $(a + 1)(b + 1) - 1$. These trends are flipped between embedding and neuron-logit map. Norms of logits in 2D Fourier basis basically follow the trend in multiplication (Figure 5[:,2]), and especially $a^2 - b$ activates key frequency columns (Figure 5[1,2]).

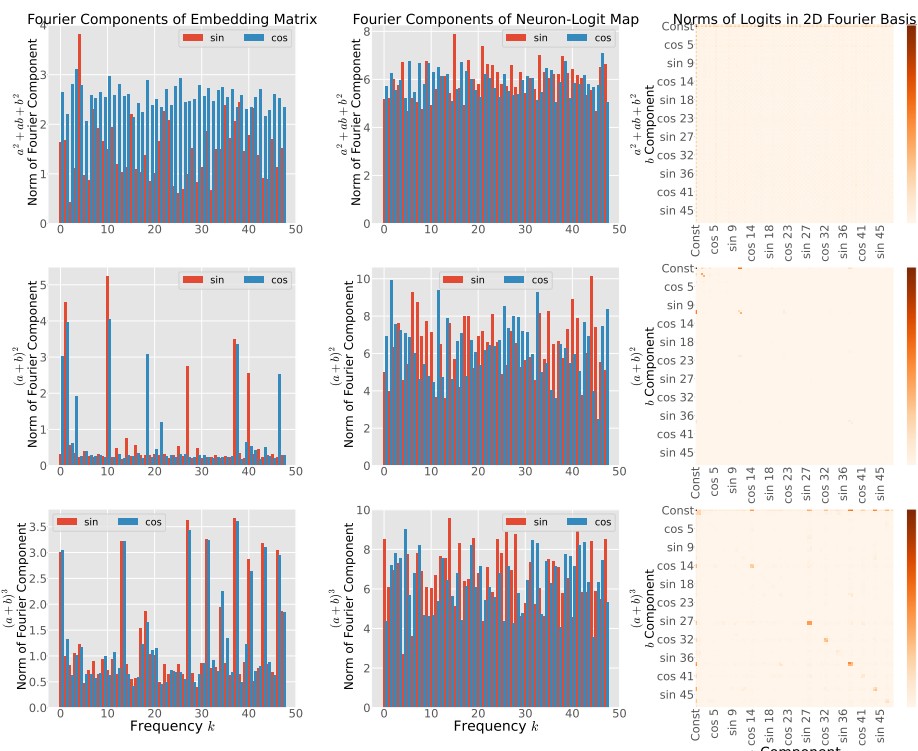

Figure 7: Frequency analysis in grokking with factorizable polynomials. Non-factorizable operation ($a^2 + ab + b^2$, fig[0, :]) cannot find sparse embedding representations. In contrast, factorization with elementary arithmetic accelerates grokking in both quadratic ($(a+b)^2$, fig[1, :]) and cubic expression ($(a+b)^3$, fig[2, :]) with sparse Fourier features.

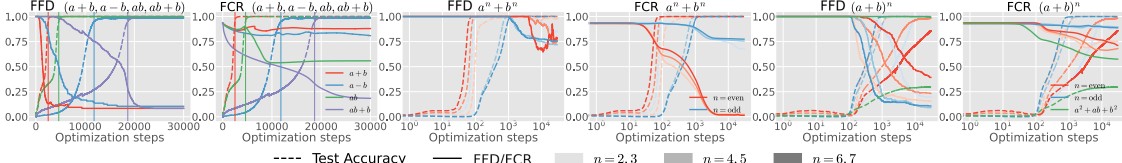

Figure 8: FFD and FCR as progress measure of grokking. The decrease of either FFD or FCR (or both) indicates the progress of grokking synchronizing with the test accuracy improvement. The responsible indicator depends on each operation. See Appendix G for the details.

## 6.2 High-Degree Factorization Allows Grokking

Increasing the complexity of operators, we test modular polynomials with quadratic, cubic, and quartic formulas in Figure 6. Apparently, Transformer fails to generalize in degree-$n$ polynomials with cross term ($a^2 + ab + b^2$, $a^2 + ab + b^2 + a$, $a^3 + ab$, $a^3 + ab^2 + b$). However, if polynomials are factorizable as a product of addition or subtraction (in the form of $(a \pm b)^n$), they easily grok, even when they also have cross terms (e.g., $(a+b)^2 + a + b$). Besides, the sum of powers is also easy to be grokked. Even, cubic (Figure 6[1, 2:4]) or quartic (Figure 6[1, 4:]) expressions, grokking occurs if they are factorizable. Comparing $a^2 + ab + b^2$ and $(a+b)^2$ or $a^2 + ab + b^2 + a$ and $(a+b)^2 + a + b$ emphasizes the importance of factorizability for the emergence of grokking.

Figure 7 analyzes the frequency components in factorizable polynomials. Non-factorizable operation ($a^2 + ab + b^2$) cannot find the sparse embedding representation. In contrast, factorizable operations promote grokking in both quadratic ($(a+b)^2$) and cubic expression ($(a+b)^3$) obtaining sparsity in embedding. The factorizable operations find more biased Fourier components than the non-factorizable ones in the neuron-logit map. Moreover, factorizable polynomials exhibit clear logits patterns as shown in elementary arithmetic (Figure 3), while non-factorizable ones only show significant norms around a constant component.

Figure 9: Test accuracy in modular linear expression $(2a \pm b, 2a \pm 3b)$ and degree-1 polynomials with cross term $(ab \pm b)$. Pre-grokked embedding in modular addition accelerates grokking in $2a \pm b, 2a \pm 3b$, and pre-grokked Transformer in modular multiplication accelerates grokking in $ab \pm b$, while the training from scratch could not generalize in $r = 0.3$.

### 6.3 FFD and FCR as Progress Measures

As shown in Figure 8, we measure FFD and FCR in embedding layer $W_E$ for various modular operations. See Appendix G for the results in neuron-logit map $W_L$.

**Elementary Arithmetic** Addition (red) and subtraction (blue) decrease FFD and keep a high FCR, whereas multiplication maintains FFD as 1.0 and decreases FCR (green). In all the cases, the saturation of accuracy and inflection point of either FFD or FCR almost match (vertical lines). Interestingly, $ab + b$ (purple) exhibits decreasing both FFD and FCR, which reflects the feature of addition and multiplication simultaneously.

**Sum of Powers** In $a^n + b^n$, FFD and FCR exhibit the same progress as multiplication, while the neuron-logit map has sparsity the same as addition (Appendix G). We also observe different behaviors depending on the parity of exponent $n$; FFD decreases more when $n$ is odd (blue) and FCR drops more when $n$ is even (red).

**Factorizable Polynomials** $(a + b)^n$ exhibits the same trend as addition: high sparsity and balanced components. In contrast, the neuron-logit map behaves similarly to multiplication (Appendix G). As in the sum of powers, the dynamics would be different depending on the parity of exponent $n$; FCR significantly drops when $n$ is even. In the case of non-factorizable $a^2 + ab + b^2$, FFD do not change during training, and the model cannot achieve late generalization.

## 7 Analysis in Transferability

Since all the modular arithmetic has periodicity, we could hypothesize that grokked models obtain common features among similar operations (transferability). Furthermore, pre-grokked models in a certain task could promote grokking in other similar tasks because they already have a useful basis. We first test the transferability of pre-grokked models from elementary arithmetic to linear expressions (Section 7.1), and then extensively investigate it with higher-order polynomials (Section 7.2).

### 7.1 Pre-Grokked Models Accelerate Grokking in Linear Expression

We test whether frozen pre-grokked modules in elementary arithmetic $(a + b, a * b)$ are transferable to grokking in modular linear expression $(2a \pm b, 2a \pm 3b, ab \pm b)$. Those asymmetric expressions are hard to grok from scratch, especially if the fraction is small $(r = 0.3)$ despite their simplicity. Figure 9 shows that pre-grokked embedding with addition accelerates grokking in $2a \pm b, 2a \pm 3b$, and pre-grokked Transformer with multiplication does in $ab \pm b$. These support our hypothesis and imply that in complex operations, internal circuits struggle with finding interpretable patterns.

### 7.2 Pre-Grokked Models May not Help Higher-Order Polynomials

In Section 7.1, we demonstrate that pre-grokked models accelerate grokking in linear expressions. We here extensively test pre-grokked models in higher-order polynomials (quadratic and cubic). Table 1 shows that pre-grokked models could not accelerate, and they even prevent grokking in higher-order polynomials, which implies that pre-grokked models may not always help grokking accelerations, except for linear expressions. While the learned representation of polynomials seems to be a superposition of that of elementary arithmetic (e.g. Section 6.1), their functionalities might differ significantly.

| Downstream Op. | Addition $(a+b)$ | | Multiplication $(a*b)$ | | Subtraction $(a-b)$ | | From Scratch |
|---|---|---|---|---|---|---|---|
| | PG-E | PG-T | PG-E | PG-T | PG-E | PG-T | |
| $2a+b$ | ✔ | ✔ | ✗ | ✔ | $r=0.4$ | $r=0.7$ | $r=0.5$ |
| $2a-b$ | ✔ | ✔ | ✗ | ✔ | ✔ | $r=0.5$ | $r=0.4$ |
| $2a+3b$ | ✔ | ✔ | ✗ | ✔ | $r=0.4$ | ✗ | $r=0.4$ |
| $2a-3b$ | ✔ | ✔ | ✗ | ✔ | $r=0.4$ | $r=0.8$ | $r=0.4$ |
| $ab+b$ | ✗ | ✔ | $r=0.4$ | ✔ | ✗ | $r=0.7$ | $r=0.5$ |
| $ab-b$ | ✗ | $r=0.4$ | $r=0.4$ | ✔ | ✗ | $r=0.7$ | $r=0.5$ |
| $(a+b)^2$ | ✔ | ✔ | $r=0.8$ | ✔ | ✔ | $r=0.9$ | ✔ |
| $(a-b)^2$ | ✔ | ✗ | $r=0.9$ | ✗ | ✔ | $r=0.8$ | ✔ |
| $(a+b)^2+a+b$ | ✔ | $r=0.4$ | ✗ | ✔ | ✔ | ✔ | ✔ |
| $a^2+ab+b^2$ | $r=0.9$ | ✗ | $r=0.7$ | ✗ | $r=0.9$ | ✗ | $r=0.8$ |
| $a^2-b$ | $r=0.4$ | ✔ | ✗ | ✔ | $r=0.6$ | $r=0.9$ | $r=0.4$ |
| $a^2-b^2$ | $r=0.6$ | $r=0.7$ | $r=0.6$ | $r=0.5$ | $r=0.7$ | $r=0.4$ | ✔ |
| $(a+b)^3$ | ✔ | ✗ | ✗ | ✗ | $r=0.6$ | ✗ | ✔ |
| $(a-b)^3$ | $r=0.4$ | ✗ | ✗ | ✗ | $r=0.6$ | ✗ | $r=0.5$ |
| $a^3+ab$ | ✗ | ✗ | ✗ | ✗ | ✗ | ✗ | $r=0.9$ |
| $a^3+ab^2+b$ | ✗ | ✗ | ✗ | ✗ | ✗ | ✗ | ✗ |

Table 1: Summary of grokked modular operators with pre-grokked models (both embedding and Transformer). We provide the smallest train fraction where grokking happens. PG-E/T stands for pre-grokked embedding/Transformer. The  shaded  ones are the results presented in Figure 9.

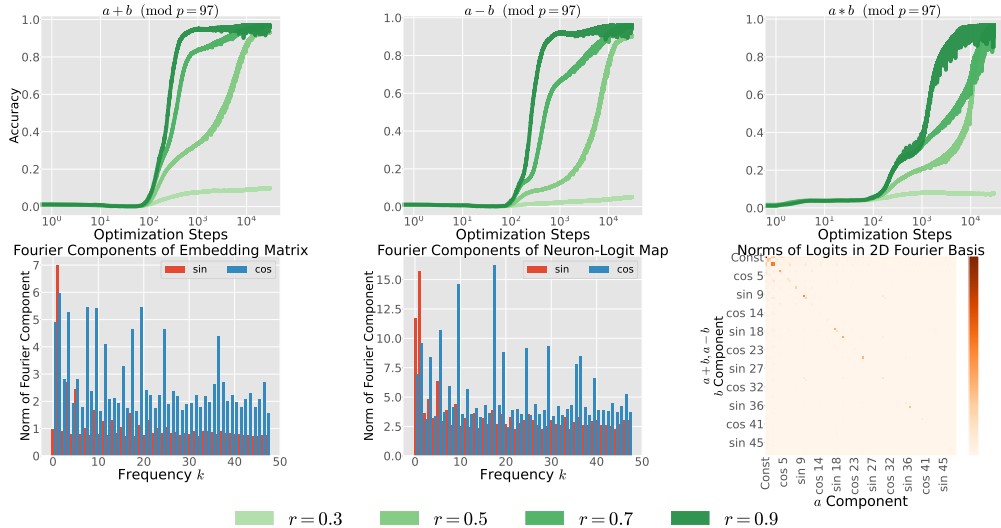

Figure 10: Test accuracy and frequency analysis in grokking with a mixture of elementary arithmetic. Co-grokking across different operations occurs, but it needs a larger fraction than a single task ($r = 0.3$ does not work).

These ablation studies reveal that the transferability of pre-grokked embeddings and models is limited to specific combinations, such as from elementary arithmetic to linear expressions, and could be rarely observed in higher-degree expressions. From the transferability of learned representation perspective, we should note that there is still an analysis gap between the grokking with synthetic data and common machine learning.

## 8 Analysis in Multi-Task Training

While previous works on grokking have only dealt with a single task during training, the application of Transformers such as large language models (Brown et al., 2020) is usually trained on a mixture of various tasks or datasets. Given the periodicity and similarity across entire modular arithmetic, we also hypothesize that mixing functionally similar operations in the dataset promotes grokking. To fill the gap between synthetic tasks and practice, we here investigate grokking on mixed datasets with addition, subtraction, and multiplication (Section 8.1). We also study multi-task training mixing hard and easy polynomial operations (Section 8.2). We prepare $r = 0.3$ datasets and jointly train Transformers on their mixture.

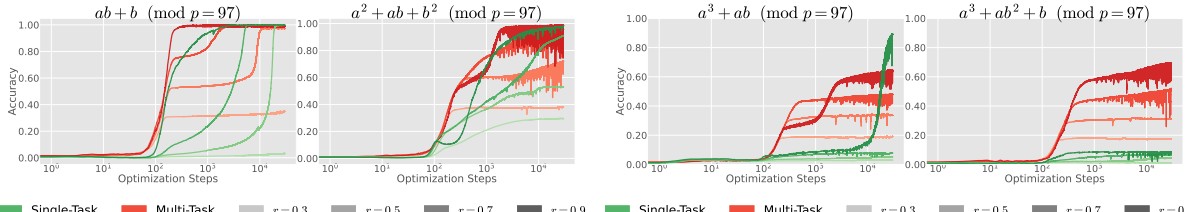

Figure 11: (**Left**) Test accuracy in grokking with a mixture of modular polynomials ($\{a + b, ab + b\}$ and $\{a^2 + b^2, a^2 + ab + b^2, (a + b)^2\}$). The multi-task training across similar operations promotes grokking. (**Right**) Test accuracy in grokking with a mixture of modular polynomials ($\{(a + b)^3, a^3 + ab\}$ and $\{(a + b)^3, a^3 + ab^2 + b\}$). The multi-task training across similar operations promotes the improvement of test accuracy.

## 8.1 Multi-Task Mixture Discovers Coexisting Solutions

Figure 10 reveals that *co-grokking* (i.e. grokking happens for all the tasks) occurs, but it requires a larger fraction of train dataset than a single task; for instance, $r = 0.3$ could not cause grokking while it does in Figure 2. The test accuracy of multiplication increases slower than the other two, which implies the conflict among different Fourier representations may affect the performance and generalization.

For the Fourier analysis of grokked models, training with a multi-task mixture seems to discover "Pareto-optimal" representations for all the operations in embedding and neuron-logit map (Figure 10[1, :]). We can see the coexistence of component sparsity in embedding (addition), asymmetric cosine sparsity in neuron-logit map (subtraction), and cosine-biased components for all the frequencies (multiplication). Furthermore, the norms of logits in 2D Fourier basis for addition and subtraction exhibit the same patterns. This means that addition and subtraction can be expressed on the same representation space originally, while they find quite different grokked models after the single-task training.

## 8.2 Proper Multi-Task Mixture also Accelerates Grokking in Polynomials

We also investigate the multi-task training with the mixture of polynomials; preparing the combination of easy and hard operations as $\{a + b, ab + b\}$, $\{a^2 + b^2, a^2 + ab + b^2, (a + b)^2\}$, $\{(a + b)^3, a^3 + ab\}$ and $\{(a + b)^3, a^3 + ab^2 + b\}$. As shown in Figure 11 (left), a proper mixture of polynomials, in terms of operation similarity, also accelerates grokking in multi-task settings. For instance, $a^2 + b^2$ and $(a + b)^2$ help generalization in $a^2 + ab + b^2$. This implies that the required representations among $\{a^2 + b^2, a^2 + ab + b^2, (a + b)^2\}$ would be the same while original single-task $a^2 + ab + b^2$ fails to grok due to the difficulty in non-factorizable cross term. The test accuracy also improves in the cubic expression (Figure 11, right). However, it hits a plateau before the perfect generalization.

The results imply that some multi-task mixtures may lead to co-grokking and accelerate generalization while others may not find optimal solutions. It would be an interesting future direction to further reveal the grokking dynamics and mechanism for multi-task training.

## 9 Conclusion

Our empirical analysis has shed light on significant differences in internal circuits and grokking dynamics across modular arithmetic. The learned representations are distinct from each other depending on the type of mathematical expressions. and despite the periodicity of modular arithmetic itself, the distinctive Fourier representations are only obtained in the operations that cause grokking. While grokking can also happen with complex synthetic data, we find that not all the insights are related to the nature seen in practical models. For instance, the ablation with frozen pre-grokked modules demonstrates that the transferability could only be limited to the specific combination of modular operations. The functional similarity between the mathematical expressions may not help. In addition, some multi-operation mixtures may lead to co-grokking and even promote generalization while others might not reach optimal solutions. We hope our extensive empirical analysis encourages the community to further bridge the gap between simple synthetic data and the data where analytical solutions are not attainable for a better understanding of grokked internal circuits.

**Limitation** We have observed all modular arithmetic operations that can cause grokking have shown interpretable trends with the Fourier basis. However, except for a few cases, we may not derive exact algorithms. It remains as future works to derive the approximate solutions covering the other modular operations beyond addition remain as future works. We also have examined a broader range of complex modular arithmetic than prior works and obtained some implications to bridge the analysis gaps between the synthetic and practical settings. However, our observations imply that the mechanism of grokking might not always share the underlying dynamics with common machine learning. Further investigations of internal circuits in practical models such as LLMs are important future directions.

### Acknowledgments

We thank Tadashi Kozuno for helpful discussion on this work. HF was supported by JSPS KAKENHI Grant Number JP22J21582.

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

## Appendix

## A  Mathematical Description of Transformer

In this section, we describe the structure of causal Transformer in our work, loosely following the notation of Elhage et al. (2021).

As defined in Section 3, we define embedding matrix as $W_E$, query, key, and value matrices of $j$-th head in the attention layer as $W_Q^j, W_K^j, W_V^j$. The input and output layer at the MLP block is denoted as $W_{\text{in}}, W_{\text{out}}$, and the unembedding matrix is denoted as $W_U$. We use ReLU for the activation functions and remove positional embedding, layer normalization, and bias terms for all the layers.

We also denote the token (one-hot representation of integers) in position $i$ as $t_i$, the initial residual stream on $i$-th token as $x_i^{(0)}$, causal attention scores from the last tokens ($t_2$, because the context length is 3) to all previous tokens at $j$-th head as $A^j$, the attention output at $j$-th head as $W_O^j$, the residual stream after the attention layer on the final token as $x^{(1)}$, the neuron activations in the MLP block as "MLP", and the final residual stream on the final token as $x^{(2)}$. "Logits" represents the logits on the final token since we only consider the loss from it.

We can formalize the logit calculation via the following equations.

- Embedding: $x_i^{(0)} = W_E t_i$
- Attention score: $A^j = \text{softmax}(x^{(0)^T} W_K^{j\,T} W_Q^j x_2^{(0)})$
- Attention block: $x^{(1)} = x_2^{(0)} + \sum_j W_O^j W_V^j (x^{(0)} A^j)$
- MLP activations: $\text{MLP} = \text{ReLU}(W_{\text{in}} x^{(1)})$
- MLP block: $x^{(2)} = W_{\text{out}} \text{MLP} + x^{(1)}$
- Logits: $W_U x^{(2)}$

Note that these focus on the operations for the representation from the final token $x_2^{(0)}$ and the above reflects the causal modeling. Following the discussion in Nanda et al. (2023), we ignore the residual connection and investigate the neuron logit map $W_L = W_U W_{\text{out}}$ as a dominant part to decide the logits.

## B    Example Python Code for Discrete Fourier Transform

In this section, we provide the example Python code to analyze the weights with discrete Fourier transform, as done in Section 5 and 6.

```python
# Import necessary libraries
import torch
import numpy as np
import pandas as pd

# Define useful functions
def to_numpy(tensor, flat=False):
    if type(tensor) != torch.Tensor:
        return tensor
    if flat:
        return tensor.flatten().detach().cpu().numpy()
    else:
        return tensor.detach().cpu().numpy()

def melt(tensor):
    arr = to_numpy(tensor)
    n = arr.ndim
    grid = np.ogrid[tuple(map(slice, arr.shape))]
    out = np.empty(arr.shape + (n+1,), dtype=np.result_type(arr.dtype, int))
    offset = 1

    for i in range(n):
        out[..., i+offset] = grid[i]
    out[..., -1+offset] = arr
    out.shape = (-1, n+1)

    df = pd.DataFrame(out, columns=['value']+[str(i)
                      for i in range(n)], dtype=float)
    return df.convert_dtypes([float]+[int]*n)

n_op = 5
p = 97
model = Transformer()

# Compute Fourier basis
fourier_basis = []
fourier_basis.append(torch.ones(p)/np.sqrt(p))
for i in range(1, p//2 +1):
    fourier_basis.append(torch.cos(2*torch.pi*torch.arange(p)*i/p))
    fourier_basis.append(torch.sin(2*torch.pi*torch.arange(p)*i/p))
    fourier_basis[-2] /= fourier_basis[-2].norm()
    fourier_basis[-1] /= fourier_basis[-1].norm()
fourier_basis = torch.stack(fourier_basis, dim=0)

# Extract the embedding weights from Transformer
W_E = model.embed.W_E[:, :-n_op]
# Extract the neuron-logit map weights from Transformer
W_out = model.blocks[0].mlp.W_out
W_U = model.unembed.W_U[:, :-n_op].T
W_L = W_U @ W_out

group_labels = {0: 'sin', 1: 'cos'}

# Appy discrete Fourier transform to embedding
fourier_embed_in = (W_E @ fourier_basis.T).norm(dim=0)
cos_sin_embed_in = torch.stack([fourier_embed_in[1::2], fourier_embed_in[2::2]])
df_in = melt(cos_sin_embed_in)
df_in['Trig'] = df_in['0'].map(lambda x: group_labels[x])
# Label the norm of Fouier components
norm_in = {'sin': df_in['value'][df_in['Trig']=='sin'], 'cos': df_in['value'][df_in['Trig']=='cos']}

# Appy discrete Fourier transform to neuron logit map
fourier_embed_out = (fourier_basis @ W_L).norm(dim=1)
cos_sin_embed_out = torch.stack([fourier_embed_out[1::2], fourier_embed_out[2::2]])
df_out = melt(cos_sin_embed_out)
df_out['Trig'] = df_out['0'].map(lambda x: group_labels[x])
# Label the norm of Fouier components
norm_out = {'sin': df_out['value'][df_out['Trig']=='sin'], 'cos': df_out['value'][df_out['Trig']=='cos']}
```

## C  Experimental Details

We summarize the hyper-parameters for the experiments (dimension in Transformers, optimizers, etc.) in Table 2. We provide the code in supplementary material.

| Name | Value |
|---|---|
| Mod $p$ | 97 |
| Epochs | 1e6 |
| Optimizer | AdamW (Loshchilov & Hutter, 2019) |
| Learning Rate | 0.001 |
| AdamW Betas | (0.9, 0.98) |
| Weight Decay $\lambda$ | 1.0 |
| Batch Size | (Full batch) |
| Max Optimization Steps | 3e5 |
| Number of Seeds | 3 |
| Embedding Dimension $d_{\text{emb}}$ | 128 |
| MLP Dimension $d_{\text{mlp}}$ | 512 |
| Number of Heads | 4 |
| Head Dimension | 32 |
| Number of Layers | 1 |
| Activation | ReLU |
| Layer Normalization | False |
| Bias Term in Weight Matrix | False |
| Weight Initialization | $\mathcal{N}(0, \frac{1}{\sqrt{d_{\text{out}}}})$ |
| Vocabulary Size $p'$ | $p + n_{\text{op}}$ (including operation tokens) |
| Context Length | 3 |

Table 2: Hyper-parameters for the grokking experiments. We follow the previous works (Power et al., 2022; Nanda et al., 2023; Zhong et al., 2023).

## D  Terminology for Mathematical Expressions

As a reference, we summarize the terminology for mathematical expressions in Table 3.

| Term | Expressions |
|---|---|
| Modular Arithmetic | $(a \circ b) \ \% \ p = c$ |
| Addition | $a + b$ |
| Subtraction | $a - b$ |
| Multiplication | $a * b$ |
| Elementary Arithmetic | all the above $(+, -, *)$ |
| Polynomials | $a^2 + b^2, a^3 + ab, (a+b)^4, \dots$ (including all the below) |
| Linear Expression (degree-1) | $2a - b, 2a + 3b, ab + b, \dots$ |
| Cross Term | $ab, ab^2, \dots$ |
| Quadratic Expression (degree-2) | $(a \pm b)^2, a^2 + ab, a^2 - b$ |
| Cubic Expression (degree-3) | $(a \pm b)^3, \dots$ |
| Quartic Expression (degree-4) | $(a \pm b)^4, \dots$ |
| Factorizable Polynomials | $(a \pm b)^n, (a \pm b)^n \pm \sum (a \pm b)^k \ (n = 2, 3, \dots, k < n)$ |
| Polynomials with Cross Term (Non-Factorizable Polynomials) | $a^2 + ab + b^2, a^3 + ab^2 + b, \dots$ |
| Sum of Powers | $a^n + b^n \ (n = 2, 3, \dots)$ |

Table 3: Terminology for mathematical expressions in this paper.

# E  Analysis of Restricted Loss in Modular Subtraction

In Figure 12, we test the restricted loss and ablated loss, the metrics proposed by Nanda et al. (2023), where the restricted loss is calculated only with the Fourier components of key frequencies, and the ablated loss is calculated by removing a certain frequency from the logits. The results show that modular subtraction has several *dependent* frequencies, which cause worse restricted loss if ablated, while they are not key frequencies (we set the threshold to $\Delta\mathcal{L} > 1e - 9$). Those dependent frequencies are not observed in modular addition. Moreover, the restricted loss for modular subtraction significantly gets worse than the original loss, which also emphasizes the subtle dependency on other frequency components.

Moreover, we extensively evaluate the relationships between loss and Fourier components. We here decompose the logits as follows:

$$\text{Logits} = (\text{Logits from key frequencies}) + (\text{Logits from non-key frequencies}) + (\text{Logits from residuals}),$$

where logits from residuals are estimated by subtracting logits of all the frequencies from the raw logits.

The results are presented in Table 4. In modular addition, we find that key frequencies contribute to the prediction and non-key frequencies only have a negligible effect on the loss (e.g. train loss v.s. ablation (d), restricted loss v.s. ablation (c)). The residuals actually hinder prediction accuracy (e.g., train loss v.s. ablation (c)). In modular subtraction, any ablations drop the performance and all the components contribute to the predictions, which implies that the grokked models in modular subtraction have informative representations to some degree over all the frequencies, even residuals in the logits.

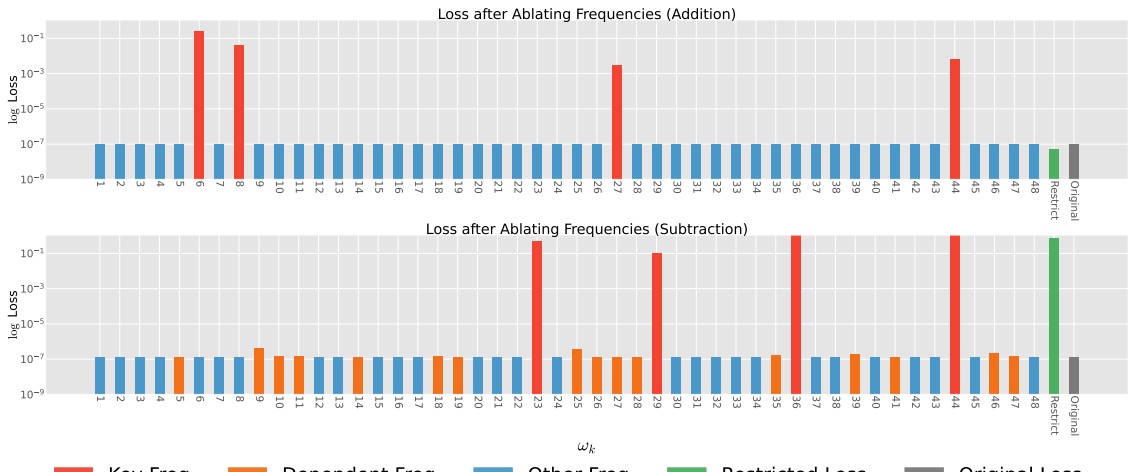

Figure 12: Loss of Transformer when ablating each frequency ($k = 1, ..., 48$) and everything except for the key frequencies (restricted loss). In modular subtraction, we find several *dependent* frequencies (orange), which cause worse restricted loss if ablated while they are not key frequencies.

|  | Logits | | | Loss ($\downarrow$) | |
|---|---|---|---|---|---|
|  | Key Freq. | Non-key Freq. | Residuals | Add ($+$) | Sub ($-$) |
| Train Loss | ✔ | ✔ | ✔ | 1.008e-7 | 1.336e-7 |
| Restricted Loss | ✔ |  |  | 4.985e-8 | 7.141e-1 |
| Ablation (a) |  | ✔ |  | 4.576 | 7.741 |
| (b) |  |  | ✔ | 5.385 | 2.179e+1 |
| (c) | ✔ | ✔ |  | 4.989e-8 | 5.582e-1 |
| (d) | ✔ |  | ✔ | 1.015e-7 | 5.348e-6 |
| (e) |  | ✔ | ✔ | 5.383 | 2.188e+1 |

Table 4: Loss of Transformer when ablating the components of key frequencies, non-key frequencies, and residuals, from the logits.

# F  Grokking with Frozen Random Embedding

We here show that even if the sparsity and non-trivial biases are not realizable in embedding, grokking could occur in Figure 13. In this experiment, we initialize embedding weights from Gaussian distribution, where the mean is 0 and the standard deviation is $\frac{1}{\sqrt{d_{\text{out}}}}$ (LeCun et al., 2012), and then freeze them not allowing any parameter updates during training. Even with the restricted capacity, the models exhibit grokking in elementary arithmetic.

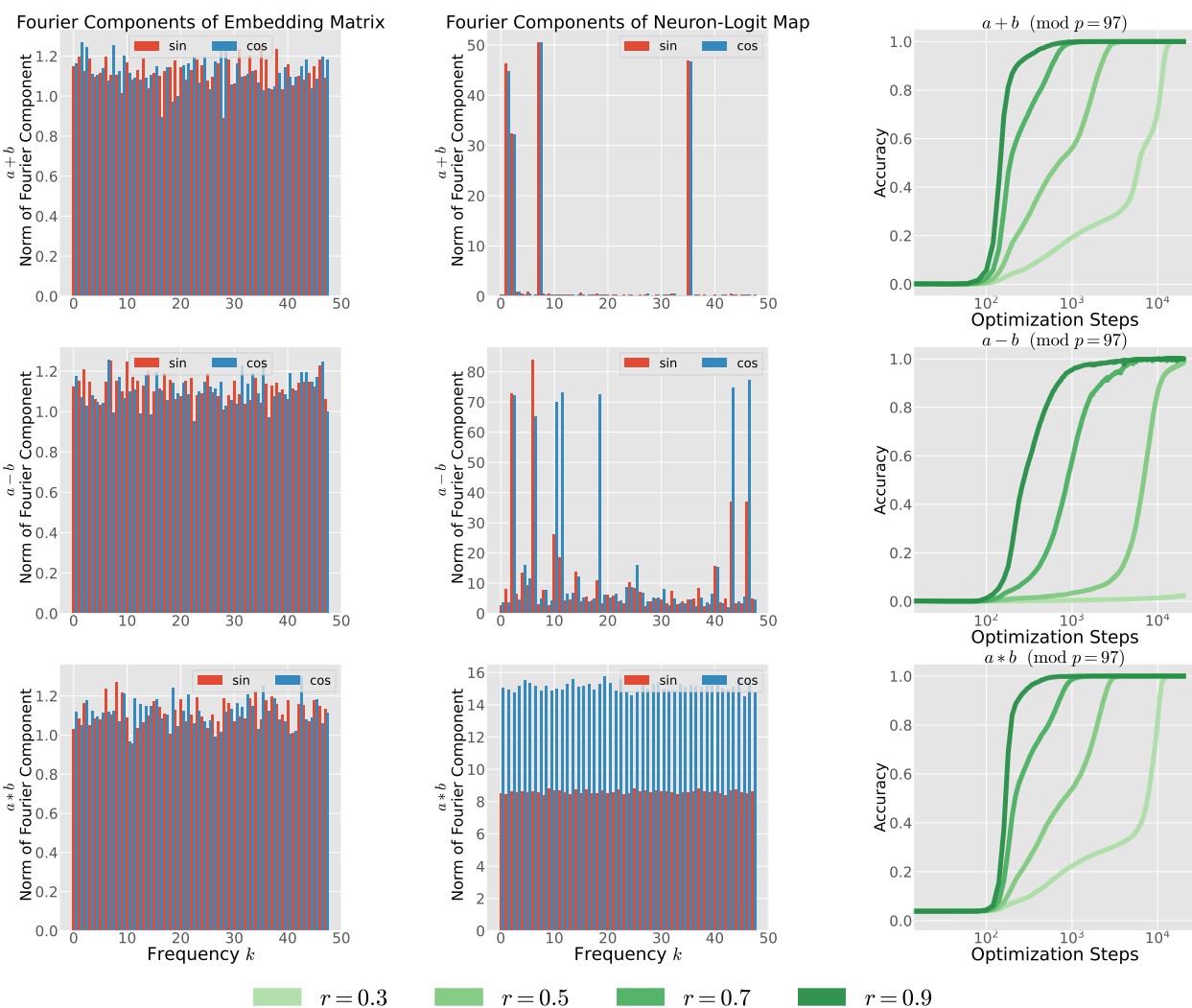

Figure 13: Test accuracy with elementary arithmetic from random embedding. Grokking can occur even using frozen random embedding, while unembedding obtains similar Fourier representation as discussed in Section 5.

## G    FFD and FCR in Neuron-Logit Map

Figure 14 presents our progress measures: FFD and FCR in neuron-logit map $W_L$. For elementary arithmetic operators, the dynamics seem to be the same as seen in embedding (Figure 8). This might be due to the similarity of embedding and neuron-logit map (Figure 3). For sum of powers ($a^n + b^n$) and the factorizable ($(a+b)^n$) behaves differently from embedding (Figure 8). The sum of powers decreases FFD while keeping FCR relatively higher. The factorizable polynomials maintain both FFD and FCR relatively higher. This might be due to the representation asymmetry between embedding and neuron-logit map in polynomials (Figure 7).

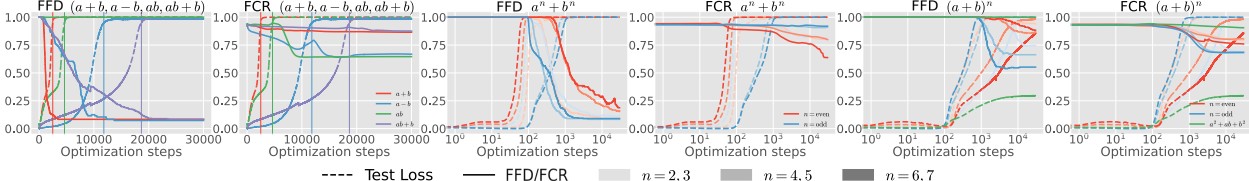

Figure 14: FFD and FCR in neuron-logit map for each operation $(a + b, a - b, a * b, ab + b, a^n + b^n, (a + b)^n)$.

## H    Detailed Analysis on FFD and FCR

In this section, we provide experiments to measure the FCR and FFD with different train dataset fractions $r$ or threshold $\eta$. As shown in Figure 15 (above), if we change the dataset ratio, the needed optimization steps to be grokked are also changed, and the inflection point of the corresponding progress measure is changed too. For instance, when we change $r = 0.3$ with $r = 0.5$ in $a + b$ and $a - b$, the decrease of FFD is also accelerated along with the increase of test accuracy. We also observe that if the grokking does not happen, the progress measure does not exhibit the change, such as the case of FFD in $ab + b$ with $r = 0.3$.

As for different threshold $\eta$, Figure 16 demonstrates that lower $\eta$ delays the decrease of the metrics in contrast to the progress of grokking; for instance, the decrease of the metrics starts after test accuracy reaches 1.0 (in the case of $a + b$, $a - b$, $ab + b$). On the other hand, larger $\eta$ results in too fast convergence. Based on those observations, we have set $\eta = 0.5$ in this paper.

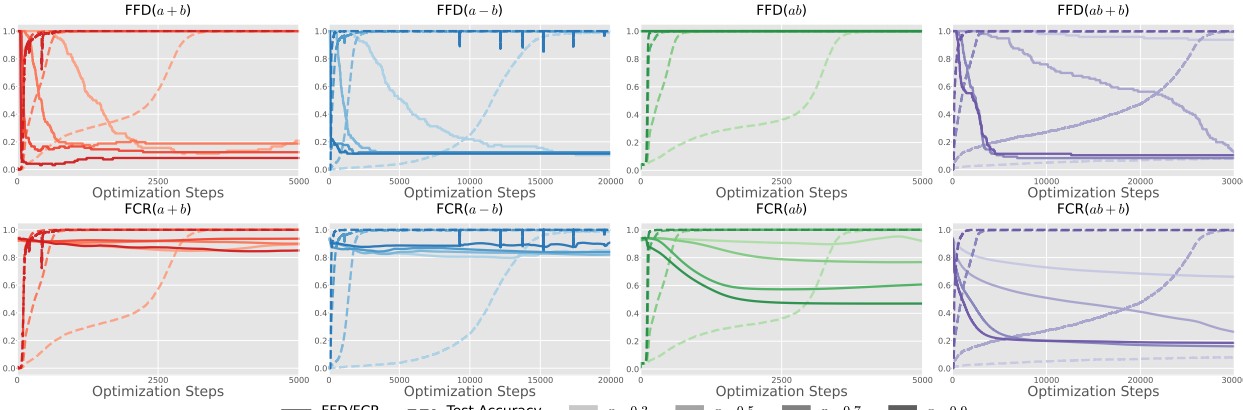

Figure 15: FFD and FCR in embedding for each operation $(a + b, a - b, a * b, ab + b)$ with different $r$.

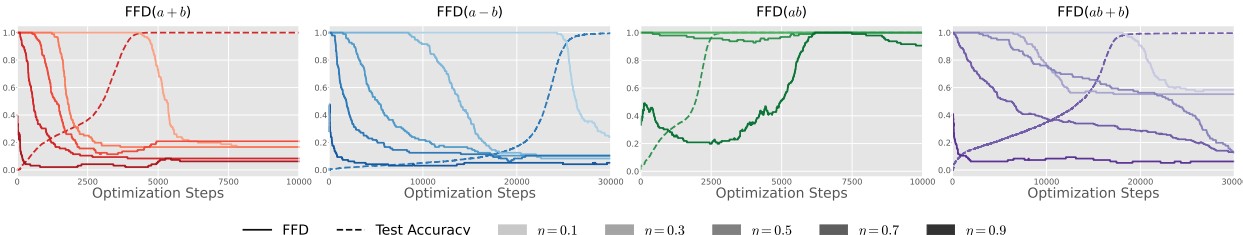

Figure 16: FFD in embedding for each operation $(a + b, a - b, a * b, ab + b)$ with different threshold $\eta$.

# I  Summary of Grokked Modular Operators

Table 5 summarizes if each modular operator can cause grokking in $r = 0.3$ or not. We provide the best test accuracy if they do not grok.

| | Elementary Arithmetic | | | Linear Expression | | | | | | | |
|---|---|---|---|---|---|---|---|---|---|---|---|
| Fraction | $a+b$ | $a-b$ | $a*b$ | $2a+b$ | $a+b \to 2a+b$ | $2a-b$ | $a+b \to 2a-b$ | $2a+3b$ | $a+b \to 2a+3b$ | $2a-3b$ | $a+b \to 2a-3b$ |
| $r = 0.3$ | ✔ | ✔ | ✔ | 3.1% | ✔ | 2.5% | ✔ | 3.3% | ✔ | 3.7% | ✔ |
| $r = 0.4$ | ✔ | ✔ | ✔ | 9.0% | ✔ | ✔ | ✔ | ✔ | ✔ | ✔ | ✔ |
| $r = 0.5$ | ✔ | ✔ | ✔ | ✔ | ✔ | ✔ | ✔ | ✔ | ✔ | ✔ | ✔ |
| $r = 0.6$ | ✔ | ✔ | ✔ | ✔ | ✔ | ✔ | ✔ | ✔ | ✔ | ✔ | ✔ |
| $r = 0.7$ | ✔ | ✔ | ✔ | ✔ | ✔ | ✔ | ✔ | ✔ | ✔ | ✔ | ✔ |
| $r = 0.8$ | ✔ | ✔ | ✔ | ✔ | ✔ | ✔ | ✔ | ✔ | ✔ | ✔ | ✔ |
| $r = 0.9$ | ✔ | ✔ | ✔ | ✔ | ✔ | ✔ | ✔ | ✔ | ✔ | ✔ | ✔ |

| | Cross Term (Degree-1) | | | | | Univariate Terms | | | | | |
|---|---|---|---|---|---|---|---|---|---|---|---|
| | $ab+a+b$ | $ab+b$ | $a*b \to ab+b$ | $ab-b$ | $a*b \to ab-b$ | $a^2+b$ | $a^2-b$ | $a^3+2b$ | $a^3-2b$ | | |
| $r = 0.3$ | ✔ | 6.1% | ✔ | 5.6% | ✔ | ✔ | 9.5% | ✔ | ✔ | | |
| $r = 0.4$ | ✔ | 9.7% | ✔ | 10% | ✔ | ✔ | ✔ | ✔ | ✔ | | |
| $r = 0.5$ | ✔ | ✔ | ✔ | ✔ | ✔ | ✔ | ✔ | ✔ | ✔ | | |
| $r = 0.6$ | ✔ | ✔ | ✔ | ✔ | ✔ | ✔ | ✔ | ✔ | ✔ | | |
| $r = 0.7$ | ✔ | ✔ | ✔ | ✔ | ✔ | ✔ | ✔ | ✔ | ✔ | | |
| $r = 0.8$ | ✔ | ✔ | ✔ | ✔ | ✔ | ✔ | ✔ | ✔ | ✔ | | |
| $r = 0.9$ | ✔ | ✔ | ✔ | ✔ | ✔ | ✔ | ✔ | ✔ | ✔ | | |

| | Cross Term (Degree-$n$) | | | | Sum of Powers | | | | | | |
|---|---|---|---|---|---|---|---|---|---|---|---|
| Fraction | $a^2+ab+b^2$ | $a^2+ab+b^2+a$ | $a^3+ab$ | $a^3+ab^2+b$ | $a^2+b^2$ | $a^2-b^2$ | $a^3+b^3$ | $a^4+b^4$ | $a^5+b^5$ | $a^6+b^6$ | $a^7+b^7$ |
| $r = 0.3$ | 34% | 4.8% | 4.9% | 4.0% | ✔ | ✔ | ✔ | ✔ | ✔ | ✔ | ✔ |
| $r = 0.4$ | 47% | 8.2% | 9.4% | 7.8% | ✔ | ✔ | ✔ | ✔ | ✔ | ✔ | ✔ |
| $r = 0.5$ | 56% | 10% | 11% | 10% | ✔ | ✔ | ✔ | ✔ | ✔ | ✔ | ✔ |
| $r = 0.6$ | 65% | 13% | 13% | 12% | ✔ | ✔ | ✔ | ✔ | ✔ | ✔ | ✔ |
| $r = 0.7$ | 74% | 17% | 14% | 13% | ✔ | ✔ | ✔ | ✔ | ✔ | ✔ | ✔ |
| $r = 0.8$ | ✔ | 42% | 16% | 15% | ✔ | ✔ | ✔ | ✔ | ✔ | ✔ | ✔ |
| $r = 0.9$ | ✔ | 67% | ✔ | 18% | ✔ | ✔ | ✔ | ✔ | ✔ | ✔ | ✔ |

| | Factorizable | | | | | | | | | | |
|---|---|---|---|---|---|---|---|---|---|---|---|
| Fraction | $(a+b)^2$ | $(a+b)^2+a+b$ | $a^2-b^2$ | $(a-b)^2$ | $(a+b)^3$ | $(a-b)^3$ | $(a+b)^4$ | $(a-b)^4$ | $(a+b)^5$ | $(a+b)^6$ | $(a+b)^7$ |
| $r = 0.3$ | ✔ | ✔ | ✔ | ✔ | ✔ | 5.9% | ✔ | 85% | ✔ | ✔ | ✔ |
| $r = 0.4$ | ✔ | ✔ | ✔ | ✔ | ✔ | 12% | ✔ | 91% | ✔ | ✔ | ✔ |
| $r = 0.5$ | ✔ | ✔ | ✔ | ✔ | ✔ | ✔ | ✔ | 91% | ✔ | ✔ | ✔ |
| $r = 0.6$ | ✔ | ✔ | ✔ | ✔ | ✔ | ✔ | ✔ | 92% | ✔ | ✔ | ✔ |
| $r = 0.7$ | ✔ | ✔ | ✔ | ✔ | ✔ | ✔ | ✔ | ✔ | ✔ | ✔ | ✔ |
| $r = 0.8$ | ✔ | ✔ | ✔ | ✔ | ✔ | ✔ | ✔ | ✔ | ✔ | ✔ | ✔ |
| $r = 0.9$ | ✔ | ✔ | ✔ | ✔ | ✔ | ✔ | ✔ | ✔ | ✔ | ✔ | ✔ |

Table 5: Summary of grokked modular operators tested in this paper ($p = 97$). We provide the best test accuracy if the operator does not cause grokking.

## J   Analysis on Grokking with Different Modulo $p$

### J.1   Periodic Patterns in Fourier Components are Common Characteristics

We here provide the ablation study with different modulo $p = 59, 113$ (from Figure 17 to Figure 21). These results show that, basically, our findings and observed trends in the main text (done with $p = 97$) can be seen in different modulo $p$ as well. The periodic patterns in Fourier components can be common characteristics among different $p$ in most cases.

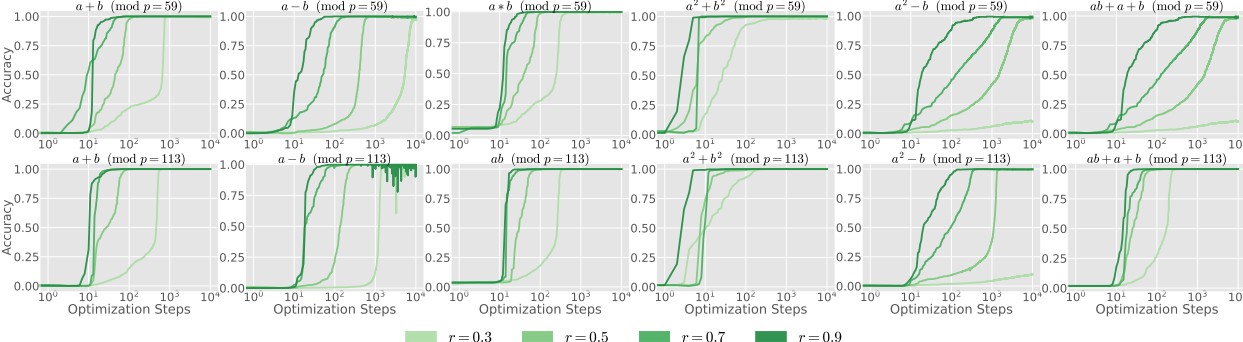

Figure 17: Test accuracy with different modulo $p \in \{59, 113\}$ $(a + b, a - b, a * b, a^2 + b^2, a^2 - b, ab + a + b)$.

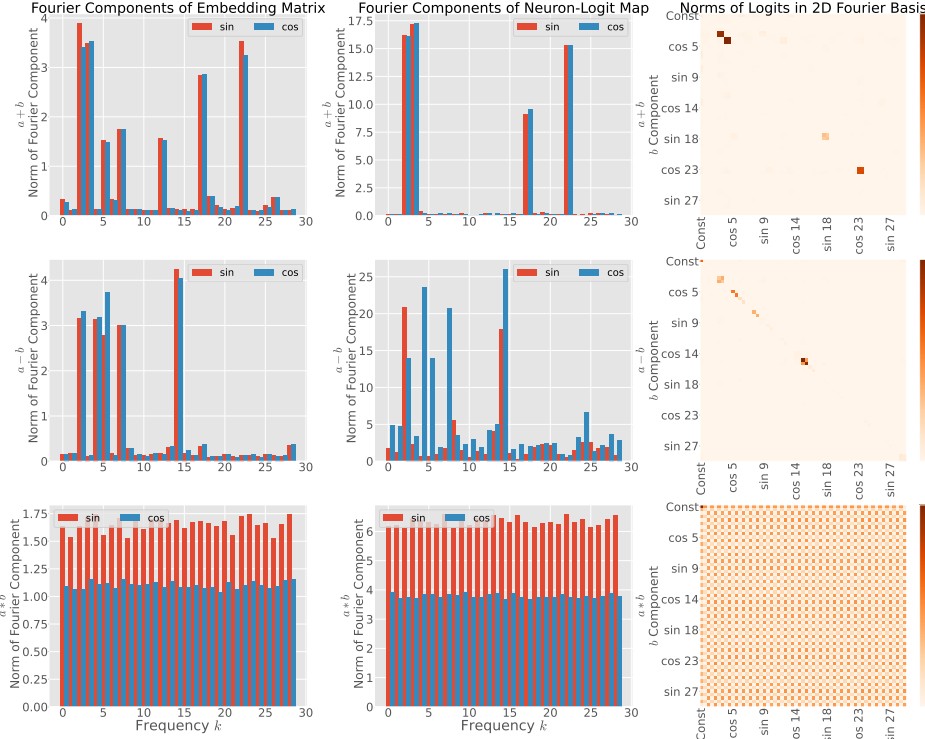

Figure 18: Frequency analysis in grokking with modulo $p = 59$ $(a + b, a - b, a * b)$.

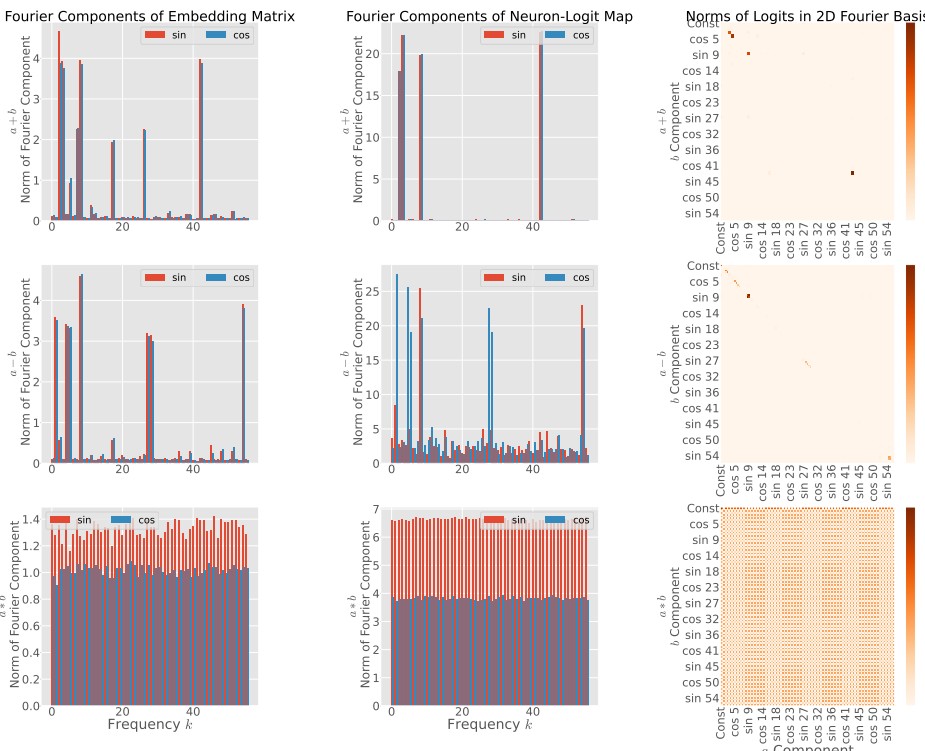

Figure 19: Frequency analysis in grokking with modulo $p = 113$ $(a + b, a - b, a * b)$.

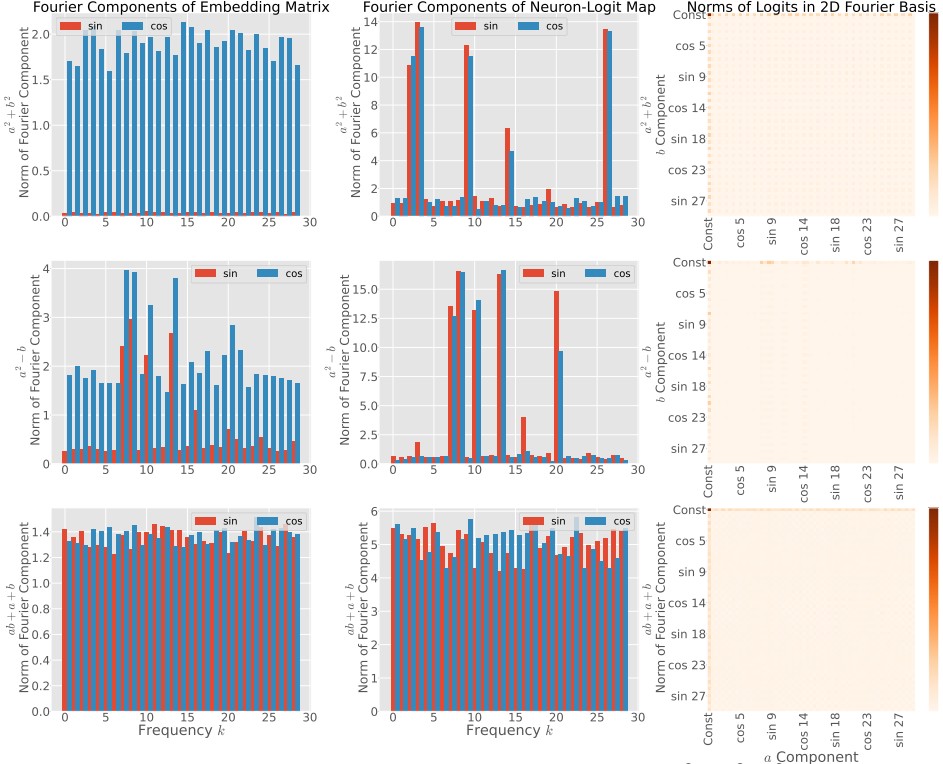

Figure 20: Frequency analysis in grokking with modulo $p = 59$ $(a^2 + b^2, a^2 - b, ab + a + b)$.

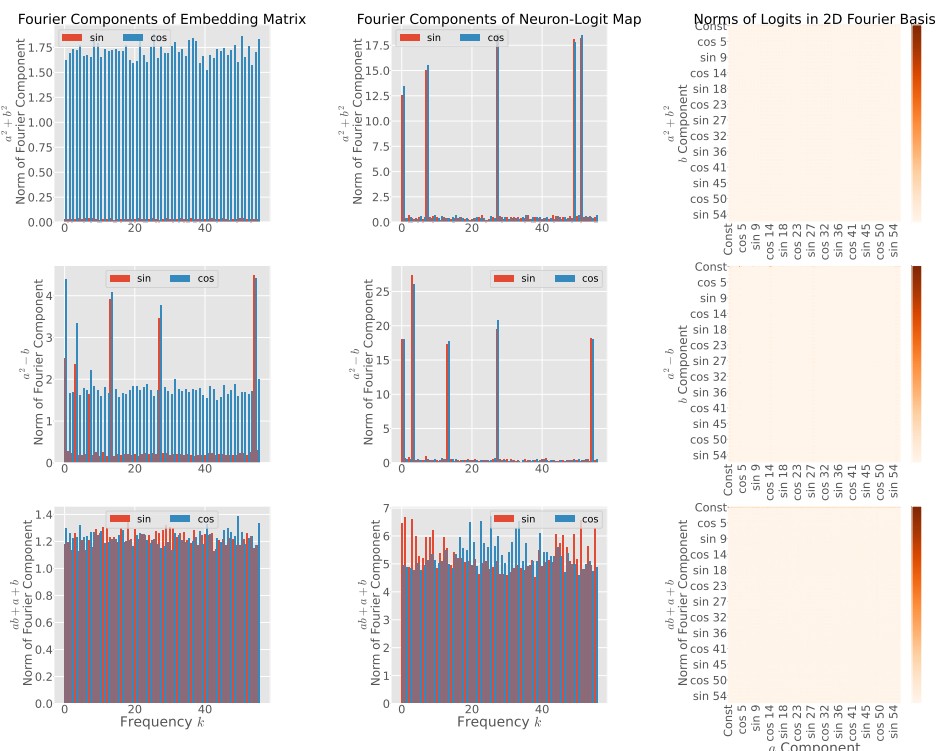

Figure 21: Frequency analysis in grokking with modulo $p = 113$ ($a^2 + b^2$, $a^2 - b$, $ab + a + b$).

## J.2 Grokking Can be a Function of Modulo $p$

In addition to mathematical operation and dataset fraction, grokking can be a function of modulo $p$. Figure 22 shows that $p = 97$ causes grokking with $a^3 + ab$, while $p = 59$ and $p = 113$ do not. Surprisingly, $p = 59$ has fewer combinations than $p = 97$, but $p = 59$ does not generalize to the test set even with $r = 0.9$. The results suggest that we might need to care about the choice of $p$ for grokking analysis.

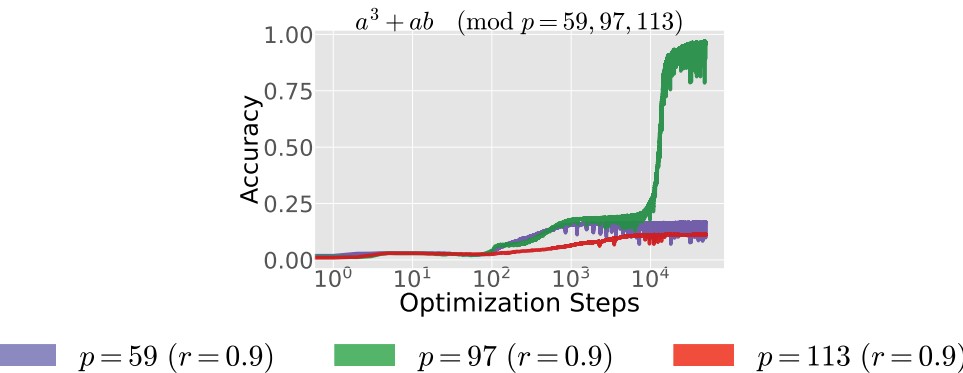

Figure 22: Test accuracy in grokking with $a^3 + ab$ ($r = 0.9$). $p = 97$ only causes grokking among $\{59, 97, 113\}$.

## K  Dataset Distribution Does not Have Significant Effects

One possible hypothesis why some modular polynomials are hard to generalize is that some polynomials bias the label distribution in the dataset. To examine this hypothesis, we calculate several statistics on label distribution in the dataset. We first randomly split train and test dataset ($r = 0.3$), and get categorical label distributions. We compute the KL divergence between train label distribution $d_{\text{train}}$ and test label distribution $d_{\text{test}}$, train label entropy, and test label entropy, averaging them with 100 random seeds.

Figure 23 shows KL divergence between train and test datasets (top), train dataset entropy (middle), and test dataset entropy (bottom). While those values slightly differ across the operations, there are no significant difference between generalizable (e.g. $a^3 + b^3$, $a^2 + b^2$) and non-generalizable (e.g. $a^3 + ab$, $a^2 + ab + b^2$) polynomials despite their similarity. The results do not imply that dataset distribution has significant impacts on grokking.

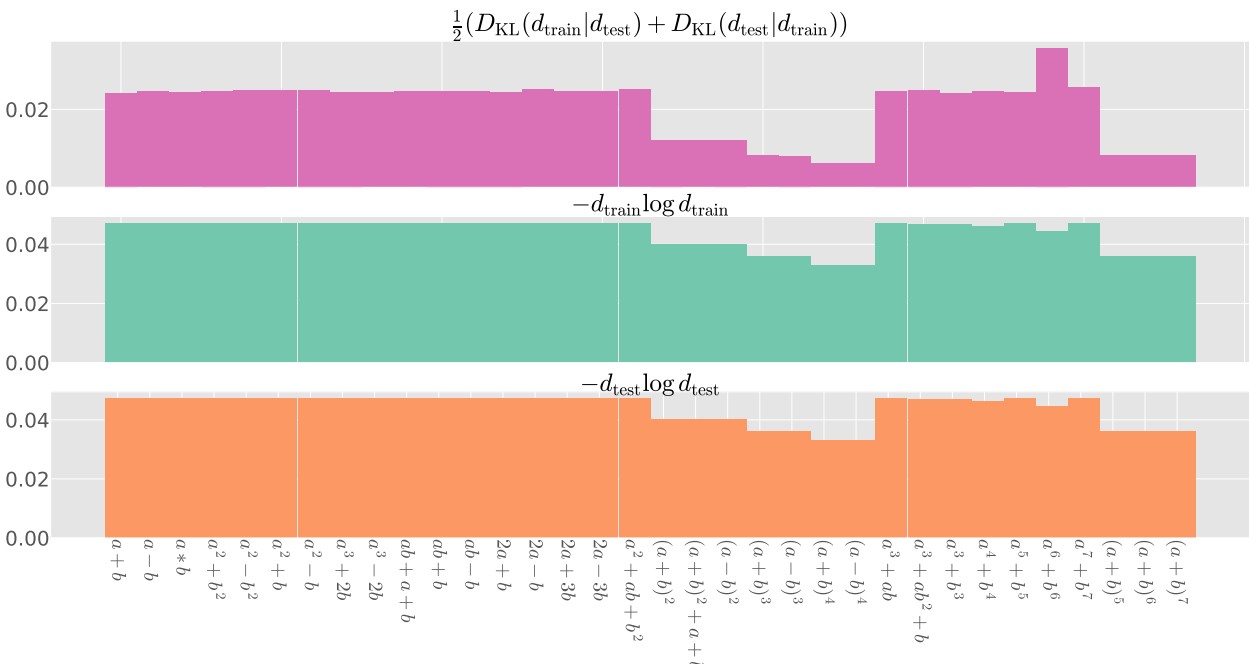

Figure 23: KL divergence between train and test datasets (top), train dataset entropy (middle), and test dataset entropy (bottom).

## L  Extended Limitation

Our work extends the grokking analysis from simple modular addition to complex modular polynomials. However, those tasks are still synthetic and far from LLMs (Brown et al., 2020), the most popular application of Transformers. Connecting grokking phenomena or mechanistic interpretability analysis into the emergent capability (Wei et al., 2022), or limitation in compositional generalization (Dziri et al., 2023; Furuta et al., 2023) and arithmetic (Lee et al., 2023) would be interesting future directions.

## M  Initialization and Pre-Grokked Models

To clarify that our experimental observations do not trivially come from random initialization, we provide a detailed version of Figure 2 in Figure 24, which has a visualization of standard deviation among three random seeds (shadow area). These results show that the overlap is small and thus our observations are consistent and do not come from the initialization or variance.

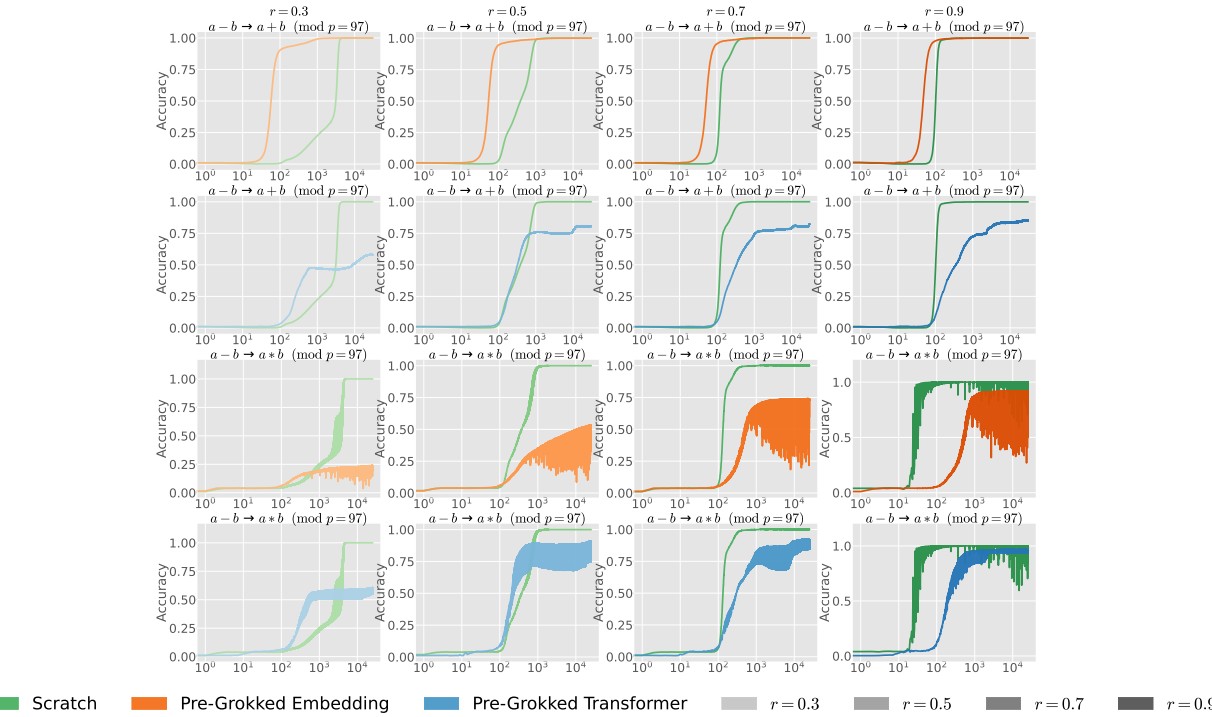

Figure 24: Test accuracy in modular elementary arithmetic (addition and multiplication) with pre-grokked models (embedding and Transformer) trained with subtraction. In addition to the plot in Figure 2, we visualize the standard deviation (shadow area).

## N  Identifiability and Pre-Grokked Models

Through the experiments with pre-grokked models, we observe that some pre-grokked models prevent grokking in certain combinations. However, Singh et al. (2024) point out that freezing a subset of models may suffer from identifiability issues, which may not prompt the downstream performance. To mitigate the potential issues, our experiments use the same random seed between pre-grokking and downstream grokking experiments. Moreover, we observe that the combinations that cannot accelerate grokking have quite different Fourier components (for instance, modular addition and multiplication as seen in Figure 3) in the weight matrix, while the rotating operations to the weight matrix do not change the Fourier basis. We think that our claims are not just due to the identifiability issues.

