# OpenReview forum: "Towards Empirical Interpretation of Internal Circuits and Properties in Grokked Transformers on Modular Polynomials"
_TMLR — Accepted by TMLR_

### Review · Reviewer_XqUP · 2024-08-28

**Summary Of Contributions:**

The authors lay out three hypotheses:
1. Any modular operations can be characterised with distinctive Fourier representations or internal circuits;
2. Grokked models obtain common features transferable among similar operations;
3. Mixing datasets with similar operations promotes grokking.

They then explore whether these hypotheses hold. In doing so, they find two progress measures for grokking: Fourier frequency sparsity and Fourier coefficient ratio. They also find that transferability is limited to specific combinations of operations, degree-$n$ polynomials sometimes do not grok, and that grokking relies on subtle frequencies within the logit domain.

**Audience:**

Yes

**Broader Impact Concerns:**

I have no reason to be concerned regarding the ethical implications of the work.

**Claims And Evidence:**

Yes

**Requested Changes:**

I have a few requested changes:
1. It might be worth renaming FFS to be Fourier Frequency Density as it seems if the number increases, the spectrum is denser?
2. I would be interested in a more rigorous statistical analysis of whether or not the progress measures actually do align with the point of grokking in the loss/accuracy. Perhaps one could vary the training ratio or another variable controlling the grokking gap in order to provide data for such a test?

**Strengths And Weaknesses:**

Generally, I think the paper provides good empirical insights into grokking that build on the existing literature. I believe it fulfils both criteria of the TMLR review process. Thus, I recommend that the paper be accepted.

General strengths:
1. I really like the empirical analysis completed in the paper. The introduction of new progress measures is always a welcome addition to the literature.
2. I think the idea of ablating certain frequencies from the logits is quite insightful and could be used in further work.
3. I like the observation that certain combinations of operations can either accelerate or inhibit grokking.
4. I really like the discussion around degree-$n$ polynomials with the cross term. I do, however, have a question about this listed under *general critiques and questions*.
5. It's interesting that co-grokking across different operations needs a larger fraction than training on a single task. I would not have necessarily expected this before reading the paper.

Some general critiques and questions:
1. I am unsure how to interpret the right-hand column of figures 3, 5, and 7. What are these providing to the paper? Perhaps they could be moved to the appendix.
2. How does the choice of $\eta$ influence the progress measures? Were there any ablations done with different values other than $0.5$?
3. Concerning grokking in degree-$n$ polynomials, did the authors consider increasing the size of the network? Or with the given amount of data, will the models always fail to generalise for some polynomials?
4. There seems to be a lack of theory as to why certain trends occur in the paper. For example, why are the identified progress measures connected to generalisation? This is not a reason to reject the paper, but it would have been nice to see!

Minor gripes:
1. I think the language could be improved. For example, in the sentence, "Specifically, they train the network using stochastic gradient descent over the cross-entropy loss L and weight decay," it is unclear who "they" refers to.
2. The use of the word "inducing" seems to be misapplied in the sentence, "weight decay is one of the key factors inducing the grokking phenomenon." Grokking can occur without weight decay, so "inducing" may not be the best choice of word here. I can provide references for this if the authors need.
3. There isn't any stated motivation for the choice of prime or the training fraction.
4. What is meant by "grokking discovers superposition of representations (frequency sparsity and bias) for elementary arithmetic"? Is this relating to superposition in the [Toy Models of Superposition](https://transformer-circuits.pub/2022/toy_model/index.html) sense of the word?

---

> ### Author Response · Authors · 2024-10-19
> **Author Response (1/2)**
>
> We thank the reviewer for careful reading and detailed feedback. We addressed your concerns and changes raised in the review.
>
> We revised the paper based on the reviewers’ comments, and the major edit was highlighted with coloring (purple). Please also check the updated manuscript.
>
> **> W1**
>
> > I am unsure how to interpret the right-hand column of figures 3, 5, and 7. What are these providing to the paper? Perhaps they could be moved to the appendix.
>
> The 2D heatmap in the right-hand column of Figure 3, Figure 5, and Figure 7 shows the norm of the 2D Fourier basis over the inputs at the logits, taking the L2 norm over the output dimension. These analyses have followed a prior work [1]. They show the periodic nature of logits in the grokked models; For instance, in modular addition, there are only a few components with significant norms, corresponding to the products of sines and cosines for the several key frequencies ($\mathrm{cos}(\omega_{k}a) * \mathrm{cos}(\omega_{k}b)$, $\mathrm{cos}(\omega_{k}a) * \mathrm{sin}(\omega_{k}b)$, $\mathrm{sin}(\omega_{k}a) * \mathrm{cos}(\omega_{k}b)$, $\mathrm{sin}(\omega_{k}a) * \mathrm{sin}(\omega_{k}b)$). In contrast, in modular subtraction, there are only a few components with significant norms, corresponding to the sines and cosines ($\mathrm{cos}(\omega_{k}a) * \mathrm{cos}(\omega_{k}b)$, $\mathrm{sin}(\omega_{k}a) * \mathrm{sin}(\omega_{k}b)$) for the several key frequencies. In modular multiplication, there are co-occurrences, corresponding to the sines components ($\mathrm{sin}(\omega_{k}a) * \mathrm{sin}(\omega_{j}b), j \in \{1, ..., (p-1)/2\}$).
>
> - [1] Nanda et al., Progress measures for grokking via mechanistic interpretability. https://arxiv.org/abs/2301.05217
>
> **> W2**
>
> > How does the choice of η influence the progress measures? Were there any ablations done with different values other than 0.5?
>
> We added the ablation of the progress measure (FFD; previous FFS) in Figure 16 (Appendix H) by changing the threshold $\eta$. The results demonstrate that lower $\eta$ delays the decrease of the metrics in contrast to the progress of grokking (an increase of test accuracy); for instance, the decrease of the metrics starts after test accuracy reaches 1.0 (in the case of $a+b$, $a-b$, $ab+b$). On the other hand, larger $\eta$ results in too fast convergence. Based on those observations, we have set $\eta=0.5$ in the main text.
>
> **> W3**
>
> > Concerning grokking in degree-n polynomials, did the authors consider increasing the size of the network? Or with the given amount of data, will the models always fail to generalise for some polynomials?
>
> In our preliminary experiments, we tested a 2-layer Transformer (compared to the current 1-layer Transformer) or larger embedding dimensions ($d=256, 512, 1024$; compared to the current $d=128$), and found that the tasks that a 1-layer Transformer/small embedding dimensions could not be grokked could not also be resolved by a 2-layer Transformer/larger embedding dimensions. We think the grokking may not depend on the size of the network.
>
> **> W4**
>
> > There seems to be a lack of theory as to why certain trends occur in the paper. For example, why are the identified progress measures connected to generalisation? This is not a reason to reject the paper, but it would have been nice to see!
>
> The invention of our progress measures (FCR, and FFD) has been based on the empirical observations that grokked models often exhibit (1) sparse Fourier components in embedding matrix (as seen in modular addition and subtraction), or (2) dense Fourier components across all the frequencies with sinusoidal biases (as seen in modular multiplication). Because grokked models in modular arithmetic often show either pattern, if we track both metrics, either metric can indicate the progress of grokking. The connection to the theoretical explanation is an ongoing, important future direction we are working on.
>
>
> **> M1**
>
> > I think the language could be improved. For example, in the sentence, "Specifically, they train the network using stochastic gradient descent over the cross-entropy loss L and weight decay," it is unclear who "they" refers to.
>
> Thank you for pointing out the writing issues. We fixed this as `prior works train the network using stochastic gradient descent over the cross-entropy loss $\mathcal{L}$ and weight decay`.

---

> ### Author Response · Authors · 2024-10-19
> **Author Response (2/2)**
>
> **> M2**
>
> > The use of the word "inducing" seems to be misapplied in the sentence, "weight decay is one of the key factors inducing the grokking phenomenon." Grokking can occur without weight decay, so "inducing" may not be the best choice of word here. I can provide references for this if the authors need.
>
> We agree that, in some cases, weight decay is not always mandatory for grokking (as mentioned in [2]), and we adjusted the tone as `The weight decay is one of the factors inducing the grokking phenomenon` (we believe, it is also true that weight decay is still necessary in many cases [3,4,5], including ours). Additionally, we mentioned the related work [2] that argues the "grokking without weight decay" in Section 2 (Related Works).
>
> - [2] Kumar et al., Grokking as the transition from lazy to rich training dynamics. https://openreview.net/forum?id=vt5mnLVIVo
>
> - [3] Power et al., Grokking: Generalization Beyond Overfitting on Small Algorithmic Datasets. https://arxiv.org/abs/2201.02177
>
> - [4] Liu et al., Omnigrok: Grokking beyond algorithmic data. https://openreview.net/forum?id=zDiHoIWa0q1
>
> - [5] Varma et al., Explaining grokking through circuit efficiency. https://arxiv.org/abs/2309.02390
>
> **> M3**
>
> > There isn't any stated motivation for the choice of prime or the training fraction.
>
> We followed [6] for the choice of prime $p=97$. We believe the choice of $p$ can be arbitrary (e.g. $p=59$ in [7], $p=113$ in [3,5], $p=67$ in [8]). For the training fraction $r$, we conducted sweeping experiments from $r=0.3$ to $r=0.9$. The summary is provided in Table 5 (Appendix I).
>
> - [6] Goromov, Grokking modular arithmetic. https://arxiv.org/abs/2301.02679
>
> - [7] Zhong et al., The Clock and the Pizza: Two Stories in Mechanistic Explanation of Neural Networks. https://arxiv.org/abs/2306.17844
>
> - [8] Minegishi et al.,Grokking Tickets: Lottery Tickets Accelerate Grokking. https://openreview.net/forum?id=WSsP7W8tqN
>
>
> **> M4**
>
> > What is meant by "grokking discovers superposition of representations (frequency sparsity and bias) for elementary arithmetic"? Is this relating to superposition in the Toy Models of Superposition sense of the word?
>
> The "superposition" here comes from the "superposition of waves" in the literature of the Fourier transformation we employed in the analysis, rather than the multiple functionalities of neurons in the "Toy Models of Superposition". Because the Fourier Components seen in modular polynomials (e.g. Figure 5) can be seen as a composition of the Fourier Components in modular addition, multiplication, or subtraction (Figure 3), we believe the usage of "superposition" here is reasonable.
>
>
> **> R1**
>
> > It might be worth renaming FFS to be Fourier Frequency Density as it seems if the number increases, the spectrum is denser?
>
> Thank you for your suggestion. We changed `Fourier Frequency Sparcity (FFS)` with `Fourier Frequency Density (FFD)` in the main text.
>
>
>
> **> R2**
>
> > I would be interested in a more rigorous statistical analysis of whether or not the progress measures actually do align with the point of grokking in the loss/accuracy. Perhaps one could vary the training ratio or another variable controlling the grokking gap in order to provide data for such a test?
>
> We provided additional experiments to measure the FCR and FFD with different train dataset fractions $r$ in Appendix H (Figure 15). The results show that (1) if we change the dataset ratio, the needed optimization steps to be grokked are also changed, and the inflection point of the corresponding progress measure is changed too.
> For instance, when we change $r=0.3$ with $r=0.5$ in $a+b$ and $a-b$, the decrease of FFD is also accelerated along with the increase of test accuracy.
> We also observe that (2) if the grokking does not happen, the progress measure does not exhibit the change (please see the case of FFD in $ab+b$ with $r=0.3$).

---

### Review · Reviewer_YGon · 2024-09-27

**Summary Of Contributions:**

The paper studies grokking behavior of training transformers to solve modular operations, including addition, subtraction and multiplication.

First, the paper finds that all the modular operations exhibit grokking in training. Second, they provide analysis on the Fourier representations within the models weights, extending the early work of Nanda et al on modular addition. Third, the paper proposes two new measures of hidden progress (based on Fourier coefficients) that seem to correlate with the occurrence of grokking. Finally, the paper also studies transferability of pre-trained embeddings and multi-task training.

**Audience:**

Yes

**Broader Impact Concerns:**

The paper does not raise ethical concerns, in my view.

**Claims And Evidence:**

Yes

**Requested Changes:**

The second part of the abstract, along with the introduction, reads vague at times. Multiple terminologies are undefined. It’s unclear what the main message is. For example:

>  superposition of the patterns from elementary arithmetic

What does this really mean?

> challenging cases

What defines easy vs hard in this setting?

> the ablation with frozen pre-grokked modules

It should say: our ablation study on the pre-grokked models reveals that …. Also, it’s unclear what transferability means here.

> co-grokking

I am pretty sure this is not a standard terminology. If it’s a new phenomenon discovered by this paper, it’s worth expanding and highlighting what it is.

> entire modular operations.

I’d replace this to be “other modular operations (beyond addition)”.

> through over a thousand experiments,

I don’t know how this is counted. It’s not necessary to make such a claim here. It perfectly suffices to just say “through extensive experiments…”

> while also investigating whether grokked models may exhibit transferability and scaling to the similarity and the number of tasks

This sentence sounds vague. Do you simply mean “whether grokked models may exhibit transferability to other tasks”? (Note that at this point of the paper, transferability hasn’t been defined.)

> End of introduction: We provide significant insights in the empirical interpretation of internal circuits learned through modular polynomials, where analytical solutions are not attainable.

How is it connected to the co-grokking stuff you were talking about?

> In contrast, we provide a detailed analysis of entire modular arithmetic, while extending the range of operations from addition to subtraction, multiplication, polynomials, and a multi-task mixture,

You can simply say, we extend the analysis from addition to ….

> weight decay is one of the key factors inducing the grokking phenomenon

Cite Varma et al here.

> Nanda et al. (2023) have first pointed out

Remove “have”

Sec 3: also specify the initialization scheme here. The Omnigrok paper, for example, suggests that the scale of initialization matters here.

> Sec 5.2:  we may not describe possible acquired algorithms for modular multiplication in a closed form, since trigonometric identities do not have multiplication formulas.

This requires some qualification. In principle, a full reverse engineering of the learned model, in the style of Nanda et al, is still conceivable. I would just say “we do not attempt to fully interp the learned model (in a closed form or pseudocode). However, we observe …”

> Section F:  In this experiment, we initialize embedding weights from Gaussian distribution

Specify the scale here.

Finally, I encourage the author(s)  the experiments for different values of p.

**Strengths And Weaknesses:**

On the positive side, the paper can be seen as a fairly comprehensive study on the internal representations of simple transformers trained on modular arithmetic tasks. This significantly extends beyond the scope of the early work of Nanda et al and Zhong et al.

The paper covers a wide range of experiments and settings. Most details are provided.

The technical sections of the paper are mostly well-written.

To my knowledge, (1) the main observations on the properties of the Fourier spectrum of learned embeddings (symmetry vs asymmetry, transferability etc), and (2) the notion of 2 measures of progress are both novel.




In terms of weakness, I find the early section (including abstract) hard to parse sometimes. I provide detailed comments below, and I do believe the paper, as it currently is, requires revision before publication.

It is somewhat underwhelming that the paper provides no attempt to fully reverse engineer a learned model on modular arithmetic (beyond addition) in an end to end fashion, similar to the prior work. While the experiments cover a wide range of settings, I find the paper lacks technical depth.

I find the two notions of progress measure a bit arbitrary. It is unclear whether they are causally responsible for grokking. Looking at the second picture of Figure 8, for example, the green line shows that the FCR measure continues to drop even when the model has fully learned to generalize.  Similarly, in the blue line, the measure almost plateaus throughout, though the accuracy goes up with a phase transition.

Finally, the paper studies p=97 only.

---

> ### Comment · Reviewer_Sx7C · 2024-10-07
> **Different values of p**
>
> I think the authors do experiment with some different values of p in Appendix I, but I agree seeing more of the results with these would be nice.

---

> ### Author Response · Authors · 2024-10-19
> **Author Response (1/3)**
>
> We appreciate the thoughtful feedback. We addressed your concerns and changes raised in the review.
>
> We revised the paper based on the reviewers’ comments, and the major edit was highlighted with coloring (purple). Please also check the updated manuscript.
>
>
> **> W1**
>
> > I find the early section (including abstract) hard to parse sometimes. I provide detailed comments below, and I do believe the paper, as it currently is, requires revision before publication.
>
> Thank you for pointing out the writing issues. We addressed these in the following response to Requested Changes (R1-R15).
>
>
> **> W2**
>
> > It is somewhat underwhelming that the paper provides no attempt to fully reverse engineer a learned model on modular arithmetic (beyond addition) in an end to end fashion, similar to the prior work. While the experiments cover a wide range of settings, I find the paper lacks technical depth.
>
> We think that end-to-end reverse engineering is a work to connect analytical explanations with empirical observations. While modular addition (as done in prior works) can write analytical solutions with trigonometric identities, this paper focuses on the empirical analysis part of modular arithmetic where such mathematical explanations have not been found so far. We think that empirical results themselves can provide insightful observations; for instance, we applied the same reverse engineering approach as modular addition, to the modular subtraction. While modular subtraction can also be explained through the trigonometric identities, the behavior of grokked models is different from the one from modular addition as described in previous works (see Section 5.1 and Appendix E).
>
> We believe that our scope in this paper is appropriately stated, and our results provide extensive empirical findings that were not observed in the grokking phenomena. The connection to the theoretical explanation is an ongoing, important future direction we are working on.
>
>
> **> W3**
>
> > I find the two notions of progress measure a bit arbitrary. It is unclear whether they are causally responsible for grokking. Looking at the second picture of Figure 8, for example, the green line shows that the FCR measure continues to drop even when the model has fully learned to generalize. Similarly, in the blue line, the measure almost plateaus throughout, though the accuracy goes up with a phase transition.
>
>
> The invention of our progress measures (FCR, and FFD) has been based on the empirical observations that grokked models often exhibit (1) sparse Fourier components in embedding matrix (as seen in modular addition and subtraction), or (2) dense Fourier components across all the frequencies with sinusoidal biases (as seen in modular multiplication). Because grokked models in modular arithmetic often show either pattern, if we track both metrics, either FCR or FFD can indicate the progress of grokking.
>
> Moreover, we think that the sensitivity of progress measures against the test accuracy might be a matter in previous works, too. For example, Nanda et al., [1] propose gini coefficients of embed matrix and neuron-logit map as a progress measure. In Figure 7 of their paper, we can see that the gini coefficient increases significantly after the test loss is mostly converged. We think identifying the best progress measure perfectly aligned with grokking still remains an open problem.
>
> Our work attempted to propose a metric that can measure the progress of grokking across any modular operations, and we think that we provided an empirical step towards rigorous analysis for progress measurement.
>
> Lastly, we provided additional ablation studies to measure the FCR and FFD with different train dataset fractions $r$ in Appendix H (Figure 15). The results show that (1) if we change the dataset ratio, the needed optimization steps to be grokked are also changed, and the inflection point of the corresponding progress measure is changed too.
> For instance, when we change $r=0.3$ with $r=0.5$ in $a+b$ and $a-b$, the decrease of FFD is also accelerated along with the increase of test accuracy.
> We also observe that (2) if the grokking does not happen, the progress measure does not exhibit the change (please see the case of FFD in $ab+b$ with $r=0.3$).
>
> - [1] Nanda et al., Progress measures for grokking via mechanistic interpretability. https://arxiv.org/abs/2301.05217
>
> **> R1**
>
> > `superposition of the patterns from elementary arithmetic` What does this really mean?
>
> The "superposition" here comes from the "superposition of waves" in the literature of the Fourier transformation we employed in the analysis because the Fourier Components seen in modular polynomials (e.g. Figure 5) can be seen as a composition of the Fourier Components in modular addition, multiplication, or subtraction (Figure 3). We fixed this as a `superposition of the Fourier components seen in elementary arithmetic`.

---

> ### Author Response · Authors · 2024-10-19
> **Author Response (2/3)**
>
> **> R2**
> > `challenging cases` What defines easy vs hard in this setting?
>
> We clarified this as `challenging non-factorizable polynomials.`
>
> **> R3**
>
> > `the ablation with frozen pre-grokked modules` It should say: our ablation study on the pre-grokked models reveals that …. Also, it’s unclear what transferability means here.
>
> We changed the sentence as you suggested. Also, we clarified transferability by explaining it as `the transferability among the models grokked with each operation`.
>
>
> **> R4**
>
> > `co-grokking` I am pretty sure this is not a standard terminology. If it’s a new phenomenon discovered by this paper, it’s worth expanding and highlighting what it is.
>
> Thank you for your suggestions. We highlighted co-grokking as `co-grokking --where grokking simultaneously happens for all the tasks --`.
>
>
> **> R5**
>
> > `entire modular operations` I’d replace this to be “other modular operations (beyond addition)”.
>
> We fixed this wording as you suggested.
>
>
> **> R6**
>
> > `through over a thousand experiments` I don’t know how this is counted. It’s not necessary to make such a claim here. It perfectly suffices to just say “through extensive experiments…”
>
> We removed this expression from the paper.
>
>
>
> **> R7**
>
> > `while also investigating whether grokked models may exhibit transferability and scaling to the similarity and the number of tasks.` This sentence sounds vague. Do you simply mean “whether grokked models may exhibit transferability to other tasks”? (Note that at this point of the paper, transferability hasn’t been defined.)
>
> Based on your suggestion, we changed this sentence as `whether grokked models may exhibit transferability among the models grokked with other operations and scaling to the similarity and the number of tasks`.
>
>
> **> R8**
>
> > End of introduction: `We provide significant insights in the empirical interpretation of internal circuits learned through modular polynomials, where analytical solutions are not attainable.` How is it connected to the co-grokking stuff you were talking about?
>
> We slightly modified this sentence as `We provide significant insights in the empirical interpretation of internal circuits learned through modular operations, where analytical solutions are not attainable` (modular polynomials --> modular operations) to include the training multiple modular arithmetics (addition, subtraction, and multiplication) at the same time (i.e. co-grokking).
>
>
>
> **> R9**
>
> > `In contrast, we provide a detailed analysis of entire modular arithmetic, while extending the range of operations from addition to subtraction, multiplication, polynomials, and a multi-task mixture` You can simply say, we extend the analysis from addition to ….
>
> We fixed this sentence as you suggested by removing redundant parts.
>
>
> **> R10**
>
> > `weight decay is one of the key factors inducing the grokking phenomenon` Cite Varma et al here.
>
> We included Varma et al., [2] here as a citation.
>
> - [2] Varma et al., Explaining grokking through circuit efficiency. https://arxiv.org/abs/2309.02390
>
>
> **> R11**
>
> > `Nanda et al. (2023) have first pointed out` Remove “have”
>
> We removed this wording as you suggested.
>
>
> **> R12 & R14**
>
> > Sec 3: also specify the initialization scheme here. The Omnigrok paper, for example, suggests that the scale of initialization matters here.
>
> > Section F: `In this experiment, we initialize embedding weights from Gaussian distribution` Specify the scale here.
>
> We initialize each weight matrix from Gaussian distribution, where the mean is 0 and the variance is $\frac{1}{\sqrt{d_{\mathrm{out}}}}$ [3] ($d_{\mathrm{out}}$ is output dimensions). We included this in Section 3, Appendix C, and Appendix F.
>
> - [3] LeCun et al., Efficient BackProp. https://link.springer.com/chapter/10.1007/978-3-642-35289-8_3
>
> **> R13**
>
> > Sec 5.2: `we may not describe possible acquired algorithms for modular multiplication in a closed form, since trigonometric identities do not have multiplication formulas.` This requires some qualification. In principle, a full reverse engineering of the learned model, in the style of Nanda et al, is still conceivable. I would just say “we do not attempt to fully interp the learned model (in a closed form or pseudocode). However, we observe …”
>
> Thank you again for your suggestions. Based on yours, we rewrite the sentence as `we do not attempt to fully interpret the learned model in a closed form or a form of pseudocode`.

---

> ### Author Response · Authors · 2024-10-19
> **Author Response (3/3)**
>
> **> W4 & R15**
>
> > Finally, I encourage the author(s) the experiments for different values of p.
>
> Thank you for your suggestion on the ablation with different modulo $p$. We added the results in Appendix J.1 (with $p=59, 113$; Figure 17-21). These results show that, basically, our findings and observed trends in the main text (done with $p=97$) can be seen in different modulo $p$ as well.
> As stated in Appendix J.2 (also in Figure 17, $ab + a + b$ (mod $p = 59$ and $p=113$, $r=0.3$)), in some cases, grokking can be a function of modulo $p$. Extending the analysis and reverse engineering across multiple modulo $p$ would be interesting future directions.

---

### Review · Reviewer_Sx7C · 2024-10-07

**Summary Of Contributions:**

The authors present an extensive analysis of grokking in various modular arithmetic functions of the form `f(a,b) = c (mod p)`. They go beyond prior work by exploring more complicated functions f (subtraction, multiplication, polynomials). Their central claims revolve around how representations from pretraining one task may enhance grokking on other tasks. They apply this methodology and find some positive transfer, indicating that common (periodic) representations are useful. They also introduce two novel progress measures (explored in Figure 8).

**Audience:**

Yes

**Claims And Evidence:**

Yes

**Requested Changes:**

(numbered for clarity)

1. (critical) Overall, I would suggest the authors strengthen the claims made in Section 5 and 6 with a more rigorous experimental methodology. Namely, they should describe the effect of initialization on timing (given how central timing is to their claims). They should also consider the identifiability/rotation issue when restoring-and-freezing pre-trained weights (another key aspect of the methodology). These are both good methodology ideas, they just need to be made rigorous.

2. (critical) Please define in the main text what you mean by pre-grokked Embedding and pre-grokked Transformer. I think the former is intuitive. The latter I'm still not clear on which matrices are being set and frozen.

3. (important) In Section 3 "Analysis in Modular Addition", the authors portray the "clock" algorithm (to use the term from Zhong et al. (2023)) as the way modular addition is implemented. This portrayal lacks the nuance that Zhong et al. (2023) introduced – specifically, that other algorithms may emerge (e.g., the "pizza"), and more importantly, not all models learn circular embeddings (which seem to be a central claim of this paper). I would strongly suggest revising this introduction to incorporate the nuance from prior work.

4. (important) In Section 4, I found the definition of FFS and FCR a bit hard to follow – namely, I would suggest the authors explicitly mathematically define the calculation of $\mu_k, \nu_k$ for a given weight matrix (e.g., $W_{emb}$).

Minor nits (MN):

1. The paper emphasizes performing over 1000 experiments. While impressive, experiments alone do not increase the significance of the work/I would suggest softening the use of this statement.
2. Page 2 first full paragraph: When introducing FFS and FCR, add abbreviations in parentheses before using (e.g., "Fourier Frequency Sparsity (FFS)").
3. Section 3, I think the formula for $r$ assumes binary tasks (two inputs), which hasn't been specified yet – I assume that's why $|S_{train}| + |S_{test}| = p^2$
4. Page 7, I would be wary of the boldness of using the word "causes" in  "addition-pre-grokked Transformer (Figure 2[0, 5]) causes grokking in multiplication." This isn't true at all values of $r$, results are shown for a single seed, and one could only claim that addition-pre-grokked *speeds up* grokking.
5. Figure 6 x-axis is a substantially different scale than Figure 2. This isn't an issue, I would just mention this in the caption since the x-tick-labels are quite small/hard to read.
6. What are the colors in Figure 16?

**Strengths And Weaknesses:**

(numbered for clarity)

# Strengths

1. The paper contains a large array of experiments across a range of modular arithmetic tasks with clear results figures and summaries (I especially enjoyed table 1 and Appendix H to be very instructive/a nice summary)
2. The insights drawn are interesting (though not fully supported – see weaknesses) and thought stimulating. Specifically, considering the usefulness of shared representations across tasks may provide an interesting angle on analogs of grokking in larger models, which are likely closer to the "multi-task training" setting.
3. Appendices are thoughtful and helpful:

      a. Appendix F is cool. It seems in line with Zhong et al. (2023)'s findings that not all models learn circular embeddings. Any intuitions as to what's going on there?

      b. I like Appendix J – it's a nice confound to check.

# Weaknesses

1. (big) In Section 5, the analysis methodology relies on the statement "if pre-grokked embedding encourages grokking in downstream tasks, the learned embedding should be similar, but if not, those tasks should require different types of representations." While the former is true, the latter isn't necessarily. Just because pre-grokked embeddings don't accelerate learning doesn't mean those tasks require different types of representations. The methodology for restoring pre-grokked embeddings involves fixing weights of a submodule (e.g., $W_{emb}$). However, this methodology suffers from the identifiability issue – namely, a rotation could be applied to all matrices. Prior work [2] has demonstrated the large effect this can have on dynamics (see Appendix D.1.1 of [2]), so I would be wary of negative claims (if grokking is discouraged, different representations are required) that rely on weight fixing (the positive claims are fine). Many of the main claims of the paper require on this negative "pre-grokked didn't help" argument.
2. (big) While the authors conduct many experiments, most conclusions revolve around grokking timing (with comparisons between timing for "from scratch" and "pre-grokked" runs being the main methodology) are only done on a single initialization. How do timings vary across initializations? Are the effects consistent, or can they be explained by this variance?
3. I would be careful with statements like "However, if polynomials are factorizable with addition (subtraction) or are the sum of powers, they easily grok, although they also have cross terms" (Section 6.2) – the results only show that the quick grokking runs are for factorizable polynomials, not the other way around? To test the provided statement, one would want to explicitly control for "factorizability". E.g., we saw that $(a^2 + b^2)$ in Figure 4 can be grokked. Does this imply that $(a+b)(a^2 + b^2)$ can always be grokked?
4. In Section 4, "The decrease of either FFS or FCR (or both) indicates the progress of grokking, and the responsible indicator depends on each modular operation" – could this statement be justified/is it obvious?
5. In Figure 2, "Addition and multiplication accelerate each other (fig[0, 5] and [2, 3])." I'm not sure if fig[0,5] actually shows this? At least at high values of $r$ from scratch seems to learn way quicker/there generally seems to be a lot of variance so I wouldn't necessarily say addition accelerates multiplication. Also, generally, it would be important to see if these results transfer across initializations for the claims being made/how much noise there is in the timing of grokking.

[1] Shalizi (2024). "Attention", "Transformers", in Neural Network "Large Language Models". http://bactra.org/notebooks/nn-attention-and-transformers.html#identification

[2] Singh et al., (2024). What needs to go right for an induction head? A mechanistic study of in-context learning circuits and their formation. https://arxiv.org/abs/2404.07129

---

> ### Author Response · Authors · 2024-10-19
> **Author Response (1/3)**
>
> We thank the reviewer for the constructive feedback. Please let us know if our responses in the following address your concerns.
>
> We revised the paper based on the reviewers’ comments, and the major edit was highlighted with coloring (purple). Please also check the updated manuscript.
>
>
> **> W1 & R1**
>
> > (big) In Section 5, the analysis methodology relies on the statement "if pre-grokked embedding encourages grokking in downstream tasks, the learned embedding should be similar, but if not, those tasks should require different types of representations." While the former is true, the latter isn't necessarily. Just because pre-grokked embeddings don't accelerate learning doesn't mean those tasks require different types of representations. The methodology for restoring pre-grokked embeddings involves fixing weights of a submodule (e.g., Wemb). However, this methodology suffers from the identifiability issue – namely, a rotation could be applied to all matrices. Prior work [2] has demonstrated the large effect this can have on dynamics (see Appendix D.1.1 of [2]), so I would be wary of negative claims (if grokking is discouraged, different representations are required) that rely on weight fixing (the positive claims are fine). Many of the main claims of the paper require on this negative "pre-grokked didn't help" argument.
>
> > (critical) Overall, I would suggest the authors strengthen the claims made in Section 5 and 6 with a more rigorous experimental methodology. Namely, they should describe the effect of initialization on timing (given how central timing is to their claims). They should also consider the identifiability/rotation issue when restoring-and-freezing pre-trained weights (another key aspect of the methodology). These are both good methodology ideas, they just need to be made rigorous.
>
>
> Thank you for pointing out the identifiability and initialization issues.
> To clarify the initialization, we would like to mention that, as written in Appendix C (Table 2), we conducted the experiments with 3 random seeds, and provided the average statistics across the seeds. For clarity, we also explicitly wrote this in Section 3.
>
> Moreover, we also included the clarification statements on the pre-grokked models as, `We also use the same random seed between pre-grokking and downstream grokking experiments to reduce identifiability issues (Singh et al., 2024)`.
> While Singh et al., [1] show the identifiability issue from different seeds may not help accelerate the emergence of an induction circuit, our experiments showed that, with some pre-grokked models, the grokking did not occur, or significantly slow down compared to "from scratch" models. We think this implies that our analysis may not only be attributed to the identifiability issues.
>
> Lastly, we avoided assertive expressions and adjusted the tone in Section 5 and 6, such as, `if pre-grokked embedding encourages grokking in downstream tasks, the learned embedding should be similar, but if pre-grokked embedding prevents grokking, those tasks may require different types of representations`. Because we also observed that the combinations that cannot accelerate grokking have quite different Fourier components (e.g. modular addition v.s multiplication as seen in Figure 3) in the weight matrix,  we think that our claims are not just due to the identifiability issues, and are appropriately supported now.
>
> - [1] Singh et al., What needs to go right for an induction head? A mechanistic study of in-context learning circuits and their formation. https://arxiv.org/abs/2404.07129
>
>
>
> **> W2**
>
> > (big) While the authors conduct many experiments, most conclusions revolve around grokking timing (with comparisons between timing for "from scratch" and "pre-grokked" runs being the main methodology) are only done on a single initialization. How do timings vary across initializations? Are the effects consistent, or can they be explained by this variance?
>
> We clarified the initialization method we employed in this paper in Section 3, Appendix C and Appendix F; we initialize each weight matrix from Gaussian distribution, where the mean is 0 and the variance is $\frac{1}{\sqrt{d_{\mathrm{out}}}}$ [2] ($d_{\mathrm{out}}$ is output dimensions). As explained in the response **W1 & R1**, we conducted the experiments with 3 random seeds, and provided the average statistics across the seeds (we also included this in Section 3). We think the effects and trends we found are consistent.
>
> - [2] LeCun et al., Efficient BackProp. https://link.springer.com/chapter/10.1007/978-3-642-35289-8_3

---

> ### Author Response · Authors · 2024-10-19
> **Author Response (2/3)**
>
> **> W3**
>
> > I would be careful with statements like "However, if polynomials are factorizable with addition (subtraction) or are the sum of powers, they easily grok, although they also have cross terms" (Section 6.2) – the results only show that the quick grokking runs are for factorizable polynomials, not the other way around? To test the provided statement, one would want to explicitly control for "factorizability". E.g., we saw that (a2+b2)  in Figure 4 can be grokked. Does this imply that (a+b)(a2+b2) can always be grokked?
>
> Thank you for pointing out the ambiguity issues. Based on our results, we fixed the sentence as `However, if polynomials are factorizable as a product of addition or subtraction (in the form of $(a \pm b)^{n}$), they easily grok, even when they also have cross terms (e.g., $(a+b)^2+a+b$). Besides, the sum of powers is also easy to be grokked.` to avoid over-claiming and confusion.
>
>
> **> W4**
>
> > In Section 4, "The decrease of either FFS or FCR (or both) indicates the progress of grokking, and the responsible indicator depends on each modular operation" – could this statement be justified/is it obvious?
>
> The proposal of our progress measures (FCR, and FFD) has been based on the empirical observations that grokked models often exhibit (1) sparse Fourier components in embedding matrix (as seen in modular addition and subtraction), or (2) dense Fourier components across all the frequencies with sinusoidal biases (as seen in modular multiplication). Because grokked models in modular arithmetic often show either pattern, if we track both metrics, either metric can indicate the progress of grokking.
>
>
>
> **> W5 & R-M4**
>
> > In Figure 2, "Addition and multiplication accelerate each other (fig[0, 5] and [2, 3])." I'm not sure if fig[0,5] actually shows this? At least at high values of r from scratch seems to learn way quicker/there generally seems to be a lot of variance so I wouldn't necessarily say addition accelerates multiplication. Also, generally, it would be important to see if these results transfer across initializations for the claims being made/how much noise there is in the timing of grokking.
> > Page 7, I would be wary of the boldness of using the word "causes" in "addition-pre-grokked Transformer (Figure 2[0, 5]) causes grokking in multiplication." This isn't true at all values of r, results are shown for a single seed, and one could only claim that addition-pre-grokked speeds up grokking.
>
> Again, as explained in the response **W1**, we conducted the experiments with 3 random seeds (originally written in Table 2, Appendix C). We think the results are consistent among different seeds.
>
> We also fixed the sentence as `With small $r$, addition and multiplication accelerate each other  (fig[0, 5] and [2, 3]).` in the caption of Figure 2.
>
>
> **> R2**
>
> > (critical) Please define in the main text what you mean by pre-grokked Embedding and pre-grokked Transformer. I think the former is intuitive. The latter I'm still not clear on which matrices are being set and frozen.
>
> We updated the description of pre-grokked Embedding and pre-grokked Transformer in Section 4; `We will use pre-grokked embedding (freezing $W_{\text{emb}}$) and Transformer (freezing all the weights except for $W_{\text{emb}} and W_{\text{U}}$) in later sections`.
>
>
> **> R3**
>
> > (important) In Section 3 "Analysis in Modular Addition", the authors portray the "clock" algorithm (to use the term from Zhong et al. (2023)) as the way modular addition is implemented. This portrayal lacks the nuance that Zhong et al. (2023) introduced – specifically, that other algorithms may emerge (e.g., the "pizza"), and more importantly, not all models learn circular embeddings (which seem to be a central claim of this paper). I would strongly suggest revising this introduction to incorporate the nuance from prior work.
>
> Thank you for pointing out the related work. To incorporate those, we added a footnote to describe the literature; `Zhong et al. (2023) revealed the existence of algorithms independent of trigonometric identities if the model does not have attention. Since we employ a Transformer with attention, this paper assumes the algorithms based on trigonometric identities.`
>
> **> R4**
>
> > (important) In Section 4, I found the definition of FFS and FCR a bit hard to follow – namely, I would suggest the authors explicitly mathematically define the calculation of μk,νk for a given weight matrix (e.g., Wemb).
>
> Based on the Fourier transformation, $\mu_{k} = \frac{2}{\left[\frac{p}{2}\right]} \sum_{t=1}^{\left[\frac{p}{2}\right]} W[t]\cos{(\omega_k t)}$, and $\nu_{k} = \frac{2}{\left[\frac{p}{2}\right]} \sum_{t=1}^{\left[\frac{p}{2}\right]} W[t]\sin{(\omega_k t)}$, where $W[t]$ is a $t$-th index of weight matrix $W$. We included this description in Section 4.

---

> ### Author Response · Authors · 2024-10-19
> **Author Response (3/3)**
>
> **> R-M1**
>
> > The paper emphasizes performing over 1000 experiments. While impressive, experiments alone do not increase the significance of the work/I would suggest softening the use of this statement.
>
> We removed this expression from the paper.
>
> **> R-M2**
>
> > Page 2 first full paragraph: When introducing FFS and FCR, add abbreviations in parentheses before using (e.g., "Fourier Frequency Sparsity (FFS)").
>
> We fixed this as you suggested.
>
>
> **> R-M3**
>
> > Section 3, I think the formula for r assumes binary tasks (two inputs), which hasn't been specified yet – I assume that's why $|S_{\mathrm{train}}|+|S_{\mathrm{test}}|=p^2$
>
> We updated the description as $\boldsymbol{x} \in [ 0, . . . , p - 1 ] \times [ 0, . . . , p - 1 ]$ in Section 3 to clarify the dataset size.
>
>
>
> **> R-M5**
>
> > Figure 6 x-axis is a substantially different scale than Figure 2. This isn't an issue, I would just mention this in the caption since the x-tick-labels are quite small/hard to read.
>
> Thank you for your suggestion. We added the caption saying `Compared to the elementary arithmetic (Figure 2), these require longer optimization steps` in Figure 6.
>
>
> **> R-M6**
>
> > What are the colors in Figure 16?
>
> We colored each bar with gradation for visibility. We fixed it using a single color now.

---

> > ### Comment · Reviewer_Sx7C · 2024-11-01
> > **Reviewer response**
> >
> > Thank you authors for addressing the points + updating the manuscript. Some questions remain, so highlighting those here:
> >
> > > the grokking did not occur, or significantly slow down compared to "from scratch" models.
> >
> > I would be careful to always use the second statement. My understanding is that grokking often occurs very late in training, so it seems difficult to justify asymptotic claims about whether or not it will occur.
> >
> > > As explained in the response W1 & R1, we conducted the experiments with 3 random seeds, and provided the average statistics across the seeds (we also included this in Section 3). We think the effects and trends we found are consistent.
> >
> > (important) Could you add some plots (perhaps to the appendix) showing the individual run timings and/or include error bars? It's hard to draw evidence from averages without a notion of variance across initialization seeds, which is what the original question was getting at.
> >
> > > Because we also observed that the combinations that cannot accelerate grokking have quite different Fourier components (e.g. modular addition v.s multiplication as seen in Figure 3) in the weight matrix, we think that our claims are not just due to the identifiability issues, and are appropriately supported now.
> >
> > This is a good point – it wasn't obvious to me at first that rotations wouldn't change the fourier basis, but I was able to use the code provided by the authors in Appendix B (with some modification) to verify this for a random embedding matrix and a random rotation matrix. Perhaps the authors could include a similar note (either as a footnote or in the appendix) for the interested reader? (minor)
> >
> > > where the mean is 0 and the variance is $\frac{1}{\sqrt{d_{out}}}$
> >
> > I suspect you mean the standard deviation is $\frac{1}{\sqrt{d_{out}}}$? Also the reference appears to argue for $\frac{1}{\sqrt{d_{in}}}$ – I know both fan-in and fan-out are used by different practitioners, so I don't doubt that using $d_{out}$ is a reasonable choice. Rather, I would just update the reference to match the claim.
> >
> > > The proposal of our progress measures (FCR, and FFD) has been based on the empirical observations that grokked models often exhibit (1) sparse Fourier components in embedding matrix (as seen in modular addition and subtraction), or (2) dense Fourier components across all the frequencies with sinusoidal biases (as seen in modular multiplication). Because grokked models in modular arithmetic often show either pattern, if we track both metrics, either metric can indicate the progress of grokking.
> >
> > Given this, for W4, could you change the wording to "The decrease of either FFS or FCR (or both) *correlates to* the progress of grokking, and the responsible indicator *varies for* each modular operation" or something like that? If it's not a causal link we should be careful.
> >
> >
> > I'd like to emphasize the most important thing: While the authors have addressed most of my points, I would want to see the empirical standard deviation of the average estimate across seeds (the authors have mentioned numerous times that they average over 3 seeds, but haven't reported any notion of variance). I'm hoping this is easy to compute and can be added to some plots, which would make the claims of the paper better justified.

---

> ### Author Response · Authors · 2024-11-08
>
> Thank you for the follow-up responses. Here are our further clarifications.
>
>
> **> Re 1**
>
> > I would be careful to always use the second statement. My understanding is that grokking often occurs very late in training, so it seems difficult to justify asymptotic claims about whether or not it will occur.
>
> To clarify this, we added a footnote saying `We optimize the models up to 3e5 gradient steps and then judge whether grokking happens or not`. The max optimization steps have been provided in Table 2 (Appendix C). In our modular arithmetic experiments, if grokking happened, it often required about 1e3 - 1e5 steps. We think 3e5 steps are sufficiently large, and such a truncation is necessary in practice.
>
>
> **> Re 2**
>
> > (important) Could you add some plots (perhaps to the appendix) showing the individual run timings and/or include error bars? It's hard to draw evidence from averages without a notion of variance across initialization seeds, which is what the original question was getting at.
>
> Picking up a few settings from Figure 2 ($a-b \rightarrow a+b$ and $a-b \rightarrow a*b$), we added test accuracy plots with standard deviation (shadow area) among 3 seeds in Appendix M (Figure 24). These results show that the overlap is small and thus our observations are consistent and do not come from the initialization or variance.
>
> **> Re 3**
>
> > This is a good point – it wasn't obvious to me at first that rotations wouldn't change the fourier basis, but I was able to use the code provided by the authors in Appendix B (with some modification) to verify this for a random embedding matrix and a random rotation matrix. Perhaps the authors could include a similar note (either as a footnote or in the appendix) for the interested reader? (minor)
>
> Thank you for your suggestion. We added Appendix N to explain the relationship among our observation, pre-grokked models, and identifiability issues.
>
>
> **> Re 4**
>
> > I suspect you mean the standard deviation is $\frac{1}{\sqrt{d_{\mathrm{out}}}}$ ? Also the reference appears to argue for $\frac{1}{\sqrt{d_{\mathrm{in}}}}$  – I know both fan-in and fan-out are used by different practitioners, so I don't doubt that using $d_{\mathrm{out}}$  is a reasonable choice. Rather, I would just update the reference to match the claim.
>
> Thank you for pointing this out. We corrected the `variance` to `standard deviation` in the latest paper.
>
> **> Re 5**
>
> > Given this, for W4, could you change the wording to "The decrease of either FFS or FCR (or both) correlates to the progress of grokking, and the responsible indicator varies for each modular operation" or something like that? If it's not a causal link we should be careful.
>
> We agree with the reviewer’s suggestion. We modified the sentence as `The decrease of either FFS or FCR (or both) correlates to the progress of grokking, and the responsible indicator varies for each modular operation` in the latest paper.

---

> > ### Comment · Reviewer_Sx7C · 2024-11-08
> > **Reviewer response**
> >
> > Thank you authors for addressing the points and adding the additional plot (Figure 24), which assuages my concerns.
> >
> > Minor nit: I think the subtitles on the new plot (Figure 24) are off for the bottom two rows (I assume those are $a-b \rightarrow a * b$ rather than $a-b \rightarrow a+b$).
> >
> > I've updated my original review to indicate that the claims are well supported, and would recommend the paper for acceptance.

---

> ### Author Response · Authors · 2024-11-09
>
> Thank you for pointing out the typo in the subtitles. We updated the manuscript with a fixed figure.

---

### Decision · Action_Editor_UP3w · 2024-11-12

**Recommendation:** Accept as is

**Comment:**

This paper expands the literature on grokking in small transformer models by considering broader set of operations (including more complex ones like polynomials), and evaluating transfer across tasks by reusing pretrained model components, as well as through multitask training. The authors identify various interesting findings in the different representation structures learned by different tasks (in the fourier basis) which relate to which tasks transfer well. Overall. the reviewers and I agree that the paper offers some interesting findings and supports its claims reasonably well. The main limitation, as with other related works, is that the focus on simple modular arithmetic tasks and small scale dataset and models necessarily limits our ability to draw more general conclusions.

**Audience:**

The reviewer agree that the results in the paper would be of interest to some of TMLR's audience.

**Claims And Evidence:**

After the revision and discussion, the reviewers agree that the paper generally supports its claims.